# Linguistic Collapse:
# Neural Collapse in (Large) Language Models

**Robert Wu**
University of Toronto, Vector Institute
`rupert@cs.toronto.edu`

**Vardan Papyan**
University of Toronto, Vector Institute
`vardan.papyan@utoronto.ca`

## Abstract

Neural collapse ($\mathcal{NC}$) is a phenomenon observed in classification tasks where top-layer representations collapse into their class means, which become equinorm, equiangular and aligned with the classifiers. These behaviours — associated with generalization and robustness — would manifest under specific conditions: models are trained towards zero loss, with noise-free labels belonging to balanced classes, which do not outnumber the model's hidden dimension. Recent studies have explored $\mathcal{NC}$ in the absence of one or more of these conditions to extend and capitalize on the associated benefits of ideal geometries. Language modelling presents a curious frontier, as *training by token prediction* constitutes a classification task where none of the conditions exist: the vocabulary is imbalanced and exceeds the embedding dimension; different tokens might correspond to similar contextual embeddings; and large language models (LLMs) in particular are typically only trained for a few epochs. This paper empirically investigates the impact of scaling the architectures and training of causal language models (CLMs) on their progression towards $\mathcal{NC}$. We find that $\mathcal{NC}$ properties that develop with scale (and regularization) are linked to generalization. Moreover, there is evidence of some relationship between $\mathcal{NC}$ and generalization independent of scale. Our work thereby underscores the generality of $\mathcal{NC}$ as it extends to the novel and more challenging setting of language modelling. Downstream, we seek to inspire further research on the phenomenon to deepen our understanding of LLMs — and neural networks at large — and improve existing architectures based on $\mathcal{NC}$-related properties. Our code is hosted on GitHub: `https://github.com/rhubarbwu/linguistic-collapse`.

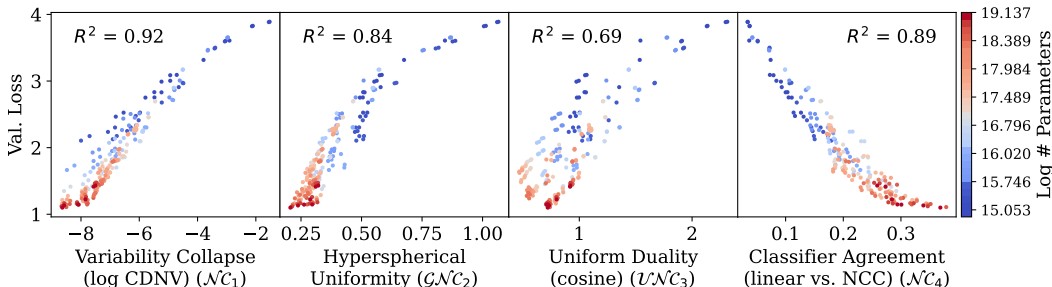

Figure 1: Simultaneous development of the four *neural collapse* ($\mathcal{NC}$) [1] properties in 230 causal language models trained on TinyStories [2], alongside improvement in generalization (i.e. validation performance). Left to right: $\mathcal{NC}_1$) within-class (representation) variability collapse; $\mathcal{GNC}_2$) hyperspherical uniformity of class means; $\mathcal{UNC}_3$) uniform duality between class means and corresponding classifiers; and $\mathcal{NC}_4$) agreement between token (maximum a prior) classifiers and implicit nearest-class centre classifiers. Coloured by model size and annotated with coefficient of determination ($R^2$).

38th Conference on Neural Information Processing Systems (NeurIPS 2024).

# 1 Introduction

## 1.1 Neural Collapse

A learning phenomenon known as *neural collapse* ($\mathcal{NC}$) emerges during the terminal phase of training (TPT) deep neural networks with cross-entropy (CE) loss for classification tasks.[1] It was originally characterized as the co-occurrence of the following properties in a model's top-layer (also known as last-layer) representations (also known as features or embeddings) and linear classifier weights:

($\mathcal{NC}_1$) **Within-class variability collapse:** Top-layer representations collapse to their class means.

($\mathcal{NC}_2$) **Convergence to a simplex ETF:** Class means tend towards equinorm and equiangular vectors when centred about the global average. The resulting geometry — known as a simplex *equiangular tight frame* (ETF) — maximizes pairwise angles and distances.

($\mathcal{NC}_3$) **Convergence to self-duality:** Linear classifier weight vectors converge to their corresponding top-layer class mean vectors, and thus also form a simplex ETF.

($\mathcal{NC}_4$) **Nearest decision rule:** Linear classifiers approximate a nearest-class centre (NCC) classifier: top-layer representations predict the class with the closest mean (implied by $\mathcal{NC}$1-3).

These behaviours, often associated with improved generalization and robustness [3–5] (among other benefits, such as those discussed in §1.4), traditionally manifest under the following conditions:

Rq1) **Few classes:** The number of classes is upper-bounded by the embedding dimension plus one: $C \leq d + 1$; this is required to construct a perfect simplex ETF.

Rq2) **Balanced classes:** The number of samples is equal across classes: $N_c = N_{c'}, \forall c \neq c'$.

Rq3) **Noise-free labels:** Identical (or very similar) embeddings should belong to the same class.

Rq4) **Sufficient training (TPT):** The model is trained past zero error, towards zero loss.

Absent these conditions, one does not typically anticipate $\mathcal{NC}$. Since then, follow-up studies have extended beyond and proposed techniques of quantifying or achieving $\mathcal{NC}$ (discussed in Section 2).

## 1.2 (Large) Language Models

$\mathcal{NC}$ is a phenomenon observed specifically in classification tasks. While not traditionally thought of as classifiers, language models — including large language models (LLMs) — learn to model aleatoric uncertainty, which can be viewed as stochastic token prediction [6]. For instance, masked language models (MLMs) such as BERT [7] predict one or several masked tokens within an input sequence based on the surrounding *context*. Likewise, autoregressive or causal language models (CLMs) such as GPT [8] perform next-token prediction (NTP) in a sequence given the *context* of previous tokens. Most of these models are essentially *(pre-)trained* by token classification on their vocabularies. This parallel — also drawn by [9] — raises a few natural questions:

1. Does the pre-training stage of a language model exhibit $\mathcal{NC}$?
2. How do scaling and other training configurations influence $\mathcal{NC}$ in (L)LMs?
3. To what extent is $\mathcal{NC}$ in (L)LMs correlated with their generalization abilities?
4. Do such correlations, between $\mathcal{NC}$ and improved generalization, persist independent of the (potential confounders of) model size and training?

To address these, we first examine the specific settings of training CLMs as they are opposed ($\neg$) to the traditional prerequisites (R1-4, §1.1) for $\mathcal{NC}$ to manifest.

¬Rq1) **Many classes:** The unique tokens found in language modelling vocabularies are vast, usually numbering in the tens of thousands and far exceeding the hidden dimension [10].

¬Rq2) **Imbalanced classes:** The distribution of tokens in natural language is typically very imbalanced [11, 12], as is the case in TinyStories [2], the dataset we use (Appendix Figure 4). It has an average of 16K samples per class but a standard deviation of over 32K.

¬Rq3) **Ambiguous contexts:** There may exist very similar or even identical contexts followed by different tokens in the natural language data [13]. For instance, over half of the sequences in TinyStories [2] lead with "`Once upon a time`", but only three-quarters of these follow with a comma ("`Once upon a time,`"). In other words, there is almost certainly some *ambiguity* to contend with in our context embeddings.

¬Rq4) **Undertraining:** Most practical language models (including LLMs) are not trained for more than a few epochs [14, 15]. Further optimization typically renders diminishing returns in improving evaluation performance [16] long before any possible TPT.

## 1.3 Contributions

We train a suite of Transformer-based [17] CLMs[1] across a grid of model widths, depths, and training epochs on the TinyStories dataset [2] to assess the degrees to which $\mathcal{NC}$ properties develop and how they relate to generalization performance. Our findings (summarized in Figure 1) reveal:

- **Emergence of $\mathcal{NC}$ with scale:** As we scale model size and training, the properties of $\mathcal{NC}$ emerge; within-class variability ($\mathcal{NC}_1$) and interference are reduced while hyperpsherical uniformity ($\mathcal{GNC}_2$), uniform duality ($\mathcal{UNC}_3$), and classifier agreement ($\mathcal{NC}_4$) improve.

- **Progression towards hyperspherical uniformity:** Class means, while unable to form a simplex ETF ($\mathcal{NC}_2$), nonetheless tend towards uniform dispersion on a hypersphere, a geometry theorized by [18] and formalized by [19] as *hyperspherical uniformity* ($\mathcal{GNC}2$).

- **Tendency towards uniform duality:** Direct alignment (self-duality) between class means and classifiers ($\mathcal{NC}_3$) does not appear to develop with scale. Instead, the variability of (mis)alignment across classes decreases with width and training, suggesting its minimization – which we term *uniform duality* ($\mathcal{UNC}_3$) – may be more cohesive with $\mathcal{NC}$.

- **Correlation between $\mathcal{NC}$ and generalization:** The developments of $\mathcal{NC}$ properties are correlated with improved validation performance. We show these correlations to persist even when fixing the (potential confounders of) model architecture and training by simply varying the random seeds for initialization and data shuffling. This suggests that $\mathcal{NC}$ is not simply a side-effect of training, but possibly a factor of generalization in its own right.

## 1.4 Significance

Recently, methods building on $\mathcal{NC}$ have found use in deep learning at large. We highlight areas such as federated learning [20], graph neural networks [21], incremental/continual learning [22–24], meta-learning [24, 25], out-of-distribution detection [26–28], privacy [29, 30], learning theory [31–36] transfer learning [3, 5, 37–40], and even LLM prompting [41]. With our results, we aim to extend insights from such contributions and other related works to the autoregressive language modelling domain and ultimately assist researchers in improving and interpreting their (L)LMs.

## 2 Related Works

$\mathcal{NC}$ was initially observed in image classification tasks such as on CIFAR-10/100 [42] and ImageNet [43]. Since then, the phenomenon has been further studied both theoretically and empirically [5, 18, 35, 44–66], with several works venturing into settings without some of the traditional prerequisites (¬Rq1-4, §1.2) and proposing adaptations of the analysis framework or optimization procedures:

**A large number of classes** (¬Rq1) $\mathcal{NC}$ established the simplex ETF as an optimal configuration. However, a perfect simplex ETF ($\mathcal{NC}_2$) requires that the number of classes $C$ not exceed $d + 1$ where $d$ is the embedding dimension. This requirement that $d$ be sufficiently large is impractical when the classes number beyond the thousands.[2] For instance, GPT-2 [68] and Llama 3.1 [69] have vocabularies of around 50K and 128K tokens, respectively.

In such a scenario, one might still expect class means to be uniformly distributed on a $d$-dimensional hypersphere [18]. [19] formalize this as *hyperspherical uniformity* within a *generalized neural*

---

[1]Our models range from 3.4M (small) to 205M (large), so we inclusively use "CLM" instead of "LLM".

[2]Following [67], one might describe such a setting ($C > d + 1$) as a model "in superposition".

*collapse* ($\mathcal{GNC}$) framework, which [9] then empirically demonstrate. These latter two works mention language modelling as applicable for $\mathcal{GNC}$; [9] even framed token prediction as a classification task, just as we do. We remark however that both earlier works *simulate* a large number of classes by drastically shrinking the embedding dimension. In contrast, we study realistic NTP, using the full class space (vocabulary) with an imbalanced token distribution and ambiguous samples.

**Class imbalance** (¬Rq2)   $\mathcal{NC}$ traditionally assumed that classes were sample-balanced. Since then, follow-up works have investigated the effect of class imbalance on the prevalence of $\mathcal{NC}$. [47] studied the *layer-peeled model* (LPM) and discovered that *minority collapse* occurs when classes are imbalanced across two equally-sized groups of classes; a threshold for minority collapse was later characterized by [35]. [52] showed that $\mathcal{NC}$ still occurs in such an LPM when the classifier is initialized as a fixed ETF. [70] introduced *simplex-encoded-label interpolation* (SELI) but noted that more severe imbalance worsens even this geometry. Recently, feature regularization has been employed to induce $\mathcal{NC}$ and improve generalization in class-imbalanced settings [58, 61, 62].

**Multi-label classification** (¬Rq3)   In some problems, one might encounter mixed or multi-label samples, be they natural or due to noise or augmentation [71, 72]. $\mathcal{NC}$ was also recently studied for such data by [73], who observed that multi-label class means arrive at the average of their labels' respective means. They also devise an augmented CE loss function to accommodate such samples.

Likewise, most of our ambiguous samples could be considered multi- or soft-label: identical (or very similar) context samples with different hard token labels (¬Rq3). Under popular CLM pre-training (teacher-forcing with CE loss), this effectively precludes the prospect of achieving zero classification error and potentially introduces irreducible noise.

A recent study showed that memorization of noisy labels likely leads to degradation of $\mathcal{NC}$ [60].

**Early stages of training** (¬Rq4)   [53] studied $\mathcal{NC}$ in small ResNet [74] models that had not yet converged (similar to most LLMs). They show that within-class variability drops and "plateaus" ($\mathcal{NC}_1$) earlier than other $\mathcal{NC}$ metrics, a result that we also observe (§4.1, Figures 6, 7).

**Natural language processing (NLP)**   An earlier study reported that the ratio of within-class to between-class covariances of word embeddings increases with model depth [75, 76], seemingly at odds with $\mathcal{NC}_1$. It can, however, be distinguished from literature studying layer-wise $\mathcal{NC}$ in that it does not centre the mean representation vectors (i.e. subtract the global mean).

[19] fine-tuned BERT [7] using their *hyperspherical uniformity gap* (HUG) objective on binary classification tasks from the GLUE benchmark [77]. [78] conducted a tangentially-related investigation of convolutional neural networks for few-class semi-supervised clustering in which they identify $\mathcal{NC}$. Our work is distinct from these in several ways: a) our class space far exceeds our embedding dimension ($C \gg d + 1$) because we classify on the full token vocabulary; b) we analyze embeddings at a token-level granularity rather than at the sequence level; and c) our NTP task is causal (context-dependent) as opposed to the per-sample independence of their few-category classification.

A more related work studied *feature collapse* in individual word representations [79], but the authors note that their analysis is limited to shallow NLP modelling on more rigid and tabular text data.

# 3 Methodology

Below we describe the training setup for our CLMs (§3.1), procedures[3] for collecting top-layer embeddings (§3.2, 3.7), and measurements of $\mathcal{NC}$ and generalization (§3.4, 3.5, 3.6, 3.7, 3.8).

## 3.1 Dataset and Language Models

TinyStories [2] is a synthetic[4] dataset generated by GPT-3.5 and GPT-4 using around 1500 English words a child might use. NTP is performed by sampling from the token vocabulary $\mathbb{V} = [\![1, 29233]\!]$,

---

[3]Leveraging our generic $\mathcal{NC}$ package: `https://github.com/rhubarbwu/neural-collapse`.

[4]TinyStories was developed and evaluated as a faithful emulation of large language modelling, so we chose it for experimentation to train hundreds of CLMs and analyze embeddings at a low cost. See Appendix A.

which for our purposes can therefore be framed as classification among $C = 29{,}233$ classes.[5] Following the GPT-style preprocessing regime [8], raw sequences are packed into $S$ chunks of size $T$, providing $N = S(T - 1)$ token samples for training.[6] Details are listed in Appendix A.

We use 30 CLM architectures based on GPT Neo [80], configured similarly to [2]. They vary in width (embedding dimension) $d \in \{64, 128, 256, 512, 768, 1024\}$ and depth (number of self-attention layers) $L \in \{1, 2, 4, 8, 12\}$. Our models were trained by teacher-forcing[7] using CE loss. For each architecture, we trained multiple models for 1, 3, and 10 epochs ablated over weight decay factors $\beta = 0.0005$ [51] and $\beta = 0.1$ [81]. Further details are listed in Appendices B, C.

## 3.2 Context Embeddings

Base CLMs perform next-token prediction: given a sequence of tokens $\boldsymbol{x}_{1:t} \in \mathbb{V}^t$, a top-layer context embedding $\boldsymbol{h}(\boldsymbol{x}_{1:t}) \in \mathbb{R}^d$ is used to predict the next token $x'_{t+1} \in \mathbb{V}$ where $1 \leq t \leq T$. A classifier for class $c$ with weights $\boldsymbol{w}_c$ and bias[8] $b_c$ would make maximum a prior (MAP) predictions as

$$x'_{t+1} := \operatorname*{argmax}_{c \in \mathbb{V}} \langle \boldsymbol{w}_c, \boldsymbol{h}(\boldsymbol{x}_{1:t}) \rangle + b_c. \tag{1}$$

**Class embedding means**  For each token class $c$, we are interested in accumulating the mean embedding $\boldsymbol{\mu}_c \in \mathbb{R}^d$ across sequences $s$ and their contexts $\boldsymbol{x}_{1:t}^{(s)}$, where the next token $x_{t+1}^{(s)}$ is ground-truth ($t < T$) and equal to $c$:

$$\boldsymbol{\mu}_c := \frac{1}{N_c} \sum_{s=1}^{S} \sum_{t=1}^{T-1} \boldsymbol{h}\left(\boldsymbol{x}_{1:t}^{(s)}\right) \mathbb{I}\left(x_{t+1}^{(s)} = c\right), \tag{2}$$

where $N_c$ is the number of samples of class $c$ and $\mathbb{I}$ is the (binary) indicator function. We also utilize their unweighted[9] average $\bar{\boldsymbol{\mu}} := \mathbb{E}_c[\boldsymbol{\mu}_c]$, and subsequently the globally-centred means $\hat{\boldsymbol{\mu}}_c = \frac{\boldsymbol{\mu}_c - \bar{\boldsymbol{\mu}}}{\|\boldsymbol{\mu}_c - \bar{\boldsymbol{\mu}}\|_2}$.

**Class embedding variances**  In a second pass, we accumulate the sample variance norms:[10]

$$\sigma_c^2 := \frac{1}{N_c} \sum_{s=1}^{S} \sum_{t=1}^{T-1} \left\| \boldsymbol{h}\left(\boldsymbol{x}_{1:t}^{(s)}\right) - \boldsymbol{\mu}_c \right\|_2^2 \mathbb{I}\left(x_{t+1}^{(s)} = c\right). \tag{3}$$

## 3.3 Homogeneity and Variability

For some collapse measures (such as $(\mathcal{G})\mathcal{NC}_2$ and $\mathcal{NC}_3$), we are primarily interested in the *variation* rather than the average of pairwise relationships. To that end, we also include in our analysis the *coefficient of variation* (CoV) of several measurements, which is the ratio of their standard deviations to their means: $\sigma(\cdot)/\mu(\cdot)$. We can interpret this as a normalized measure of variability.

## 3.4 Signal-to-Noise Ratio — $\mathcal{NC}1$

The ability to disambiguate between classes depends on the ratio of within-class to between-class variabilities. Building upon foundational works [85, 86], $\mathcal{NC}$ originally measured variability through an inverse *signal-to-no ratio* (SNR), whose minimization constitutes *within-class variability collapse* ($\mathcal{NC}_1$). We instead employ a *class-distance normalized variance* (CDNV) similar to [3]:

$$\hat{\sigma}_{c,c'} := \frac{1}{\|\boldsymbol{\mu}_c - \boldsymbol{\mu}_{c'}\|_2^k} \cdot \frac{\sigma_c^2 + \sigma_{c'}^2}{2\|\boldsymbol{\mu}_c - \boldsymbol{\mu}_{c'}\|_2^2}, \quad \forall c \neq c'. \tag{4}$$

Our metric differs in that we divide by an additional power $\|\boldsymbol{\mu}_c - \boldsymbol{\mu}_{c'}\|_2^k$ of the mean distance norm. This further downweights the CDNV within well-separated class pairs in favour of emphasizing more

---

[5]Although the GPT-Neo [80] tokenizer has over 50K tokens, only a subset vocabulary appears in TinyStories.

[6]We cannot use the first ground-truth nor the last predicted token in any chunk.

[7]Parallelized training using sequences' ground-truth labels for context as opposed to predicted tokens.

[8]Similar to many causal LLMs [80–84], our classifiers do not include additive biases, so $b_c = 0$.

[9]Different from most literature where classes were balanced and weighting is already equal.

[10]This sample variance is computed across all unnormalized dimensional entries.

mutually noisy pairs. We found this augmented CDNV with $k = 2$ to be especially useful in our setting of many imbalanced and variably confusable token classes.

These pairwise measures constitute the off-diagonal[11] entries of a symmetric matrix in $\mathbb{R}^{C \times C}$, whose average we use as an inverse SNR. Within-class variability collapse is then re-characterized by the minimization of this quantity: $\hat{\sigma}_{c,c'} \to 0, \forall c \neq c'$. This alternative convergence is empirically faithful to $\mathcal{NC}_1$ but more robust and numerically stable [3].

### 3.5 Geometric Structures — $(\mathcal{G})\mathcal{NC}2$

The separability of our representations also depends on the geometric structures found in our embeddings. [1] characterize $\mathcal{NC}_2$ as convergence to a *simplex equiangular tight frame* (ETF) [87, 88].

**Equinormness**   Such a near-orthonormal configuration firstly implies that class means are equinorm:
$$\log \|\boldsymbol{\mu}_c - \bar{\boldsymbol{\mu}}\|_2 - \log \|\boldsymbol{\mu}_{c'} - \bar{\boldsymbol{\mu}}\|_2 \to 0, \quad \forall c \neq c'. \tag{5}$$
We measure CoV in the *logarithms* of class mean norms to assess the degree of "equinormness".

**Equiangularity**   $\mathcal{NC}_2$ also entails that class means are equiangular about their centre $\bar{\mu}$: pairwise distances and angles between their class means should be maximized and similar. Following [1], we measure *interference* (sometimes known as similarity or *coherence* [89, 90]). Its minimization,
$$\langle \hat{\boldsymbol{\mu}}_c, \hat{\boldsymbol{\mu}}_{c'} \rangle \to \frac{-1}{C - 1}, \quad \forall c \neq c', \tag{6}$$
together with equinormness (Equation 5) constitute convergence to a simplex ETF. Although this geometry is not ultimately attainable since there are too many classes ($C > d + 1$), it can still be meaningful to measure a model's tendency towards one. As with CDNV noise (Equation 4), pairwise interferences form off-diagonal[12] entries in a symmetric matrix in $\mathbb{R}^{C \times C}$. The minimization of CoV in interferences therefore expresses the degree of "equiangularity".

**Hyperspherical uniformity**   A relaxation from the ETF is *hyperspherical uniformity* ($\mathcal{GNC}_2$), with equinorm (Eq. 5) means $\boldsymbol{\mu}_c$ uniformly distributed on the $d$-dimensional hypersphere [18, 19]. We likewise gauge the angular uniformity with pairwise interactions through some distance kernel $K$:[13]
$$\sum_{c \neq c'} K\left(\hat{\boldsymbol{\mu}}_c, \hat{\boldsymbol{\mu}}_{c'}\right) \to \min_{\hat{\boldsymbol{\mu}}_1, \ldots, \hat{\boldsymbol{\mu}}_C} \sum_{c \neq c'} K\left(\hat{\boldsymbol{\mu}}_c, \hat{\boldsymbol{\mu}}_{c'}\right). \tag{7}$$

### 3.6 Alignment Between Classifiers and Class Embedding Means — $(\mathcal{U})\mathcal{NC}3$

The linear classifiers $\{\boldsymbol{w}_c\}_{c=1}^C$ lie in a dual vector space to that of the class means $\{\boldsymbol{\mu}_c\}_{c=1}^C$. While convergence to self-duality ($\mathcal{NC}_3$) was initially measured as distances [1] between class means and classifiers (Equation 11), we follow [5] to inspect class-wise cosine similarities:[14]
$$\left\langle \frac{\boldsymbol{w}_c}{\|\boldsymbol{w}_c\|_2}, \hat{\boldsymbol{\mu}}_c \right\rangle \to 1, \quad \forall c \in \mathbb{V}. \tag{8}$$

For intuition analogous to that for equinormness and equiangularity (§3.5), we also measure *uniform duality* ($\mathcal{UNC}_3$) as the minimization of the CoV of these similarities (Appendices N, O).

### 3.7 Agreement of the Classifiers — $\mathcal{NC}4$

Finally, $\mathcal{NC}_4$ is described as the simplification (or approximation) of the linear classifier's MAP prediction behaviour (Equation 1) to that of the implicit *nearest-class centre* (NCC) classifier:
$$\underset{c \in \mathbb{V}}{\arg\max} \langle \boldsymbol{w}_c, \boldsymbol{h} \rangle + b_c \to \underset{c \in \mathbb{V}}{\arg\min} \|\boldsymbol{h} - \boldsymbol{\mu}_c\|_2, \quad \forall \boldsymbol{h}. \tag{9}$$

---

[11]The main diagonal of CDNVs is undefined (due to the zero denominator) and ignored.

[12]The main diagonal of interferences is equal to $\mathbf{1}$ (maximal coherence or self-similarity).

[13]Following [19], we employ the logarithmic inverse distance kernel $K_{\log}(\boldsymbol{a}, \boldsymbol{b}) = \log \|\boldsymbol{a} - \boldsymbol{b}\|_2^{-1}$ for its ability to emphasize gaps between small distances while scaling down the effect of larger distances.

[14]Dot-product would be confounded by norms and therefore inappropriate for similarity up to rescaling.

We calculate[15] agreement as the proportion of validation samples on which the classifiers agree:[16]

$$\frac{1}{N_{\text{val}}} \sum_{s=1}^{S_{\text{val}}} \sum_{t=1}^{T_{\text{val}}-1} \mathbb{I}\left( x'_{t+1}{}^{(s)} = \underset{c \in \mathcal{V}}{\arg\min} \left\| \boldsymbol{h}\left( x_{1:t}^{(s)} \right) - \boldsymbol{\mu_c} \right\|_2 \right). \tag{10}$$

### 3.8  Probing a Relationship Between $\mathcal{NC}$ and Generalization

To isolate the effect of $\mathcal{NC}$ on generalization independent of model scaling and training (if it exists), we selected a two-layer 768-wide architecture of which to train twenty more instances with weight decay $\beta = 0.0005$, each with a different data shuffling seed. We then followed the remainder of the pipeline described above to collect and analyze embeddings with respect to $\mathcal{NC}$. Finally, we performed a permutation test with $10^4$ trials to determine the statistical significance of any correlation between $\mathcal{NC}$ and generalization that remains when we hold constant all factors but shuffling seeds.

## 4  Experimental Results

In this section, we present the results from our empirical study on scaling and generalization:

($\mathcal{NC}_1$) Within-class variability is reduced across model scale (more so by width than depth) and training (up to 6 epochs), and is tightly correlated with validation performance.

($\mathcal{NC}_2$) Equinormness/equiangularity shows some improvement with scale, training, and performance. Hyperspherical uniformity ($\mathcal{GNC}_2$) also improves but more clearly and consistently.

($\mathcal{NC}_3$) Our models fail to achieve self-duality: class means do not align with classifiers. However, uniform duality ($\mathcal{UNC}3$) is correlated with model width, training, and performance.

($\mathcal{NC}_4$) Larger or more trained models exhibit closer agreement between their linear and implicit NCC classifiers. Agreement is also associated with validation performance.

### 4.1  Within-Class Variability Collapse — $\mathcal{NC}1$

Scaling our models dramatically reduces normalized variance, which is further aided by more training epochs and stronger weight decay (Appendix Figs. 6, 7). These noise reductions tightly associate with generalization (Fig. 1, left, "$\mathcal{NC}_1$"). The relationship is most apparent at scale.

### 4.2  Geometric Structures — $(\mathcal{G})\mathcal{NC}2$

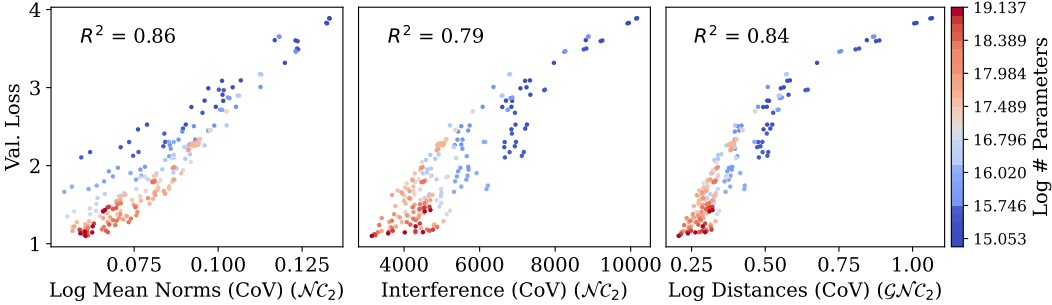

Figure 2: Validation loss is correlated with all three measurements: **(left)** equinormness ($\mathcal{NC}2$) expressed as variation in logarithmic norms; **(centre)** equiangularity ($\mathcal{NC}2$) as variation in interference; **(right)** hyperspherical uniformity ($\mathcal{GNC}2$) as variation in logarithmic pairwise distances.

**Equinormness**  Logarithmic class mean norms grow with model width and training (Appendix Fig. 8), and subtly with depth (Appendix Fig. 9). Meanwhile, the variation of these (logarithmic) norms consistently decreases (improving equinormness) with scale (Appendix Figs. 10, 11). Both trends correlate with improved generalization (Appendix Fig. 20).

---

[15]In practice, we use an equivalent decomposition (Eq. 12).

[16]Note that agreement is not equivalent to accuracy.

**Equiangularity**   Scaling model dimensions reduces average interference (Appendix Figs. 12, 13) down to an empirical plateau of approximately $10^{-3}$, which is more apparent in less-trained models. However, the variation of interference rises and persists when scaling (Appendix Figs. 14, 15), suggesting that interference becomes more concentrated between some pairs of classes. These results could be due to various factors, including but not limited to unfriendly conditions of language modelling (§1.2) or the impossibility of a perfect simplex ETF (§3.5).

Appendix Figure 21 shows only a rough performance correlation with average interference (i.e. coherence) and a more definite — albeit still noisy — one with the variation of interference (i.e. equiangularity). In other words, the limited trends we observed toward a simplex ETF suggest that the association of $\mathcal{NC}2$ with generalization begins to break down when $C > d + 1$.

**Hyperspherical uniformity**   Logarithmic distances drop more gradually and consistently with scale (Appendix Figs. 16, 17), implying this quantity is more robust or may not face the same geometric barriers seen in conventional interference (Appendix Figs. 14, 15). Variation of these logarithmic distances is also consistently reduced with scale (Appendix Fig. 18, 19).

And finally, generalization has much stronger correlations with logarithmic distances than it has with regular interference (Fig. 2), validating the applicability of $\mathcal{GNC}$ [19] when $C > d + 1$.

### 4.3   Classifier (Mis)alignment and Duality — $(\mathcal{U})\mathcal{NC}3$

Model width ($d$) is correlated faintly with the average similarity between class means and their respective classifiers (Appendix Fig. 23), but strongly with variation in similarity (Appendix Fig. 25). The relationships to generalization follow the same pattern (Fig. 3), suggesting that uniform duality ($\mathcal{UNC}_3$) *might* serve as a better $\mathcal{NC}$ property than self-duality ($\mathcal{NC}_3$) overall; we discuss this in §5.1.

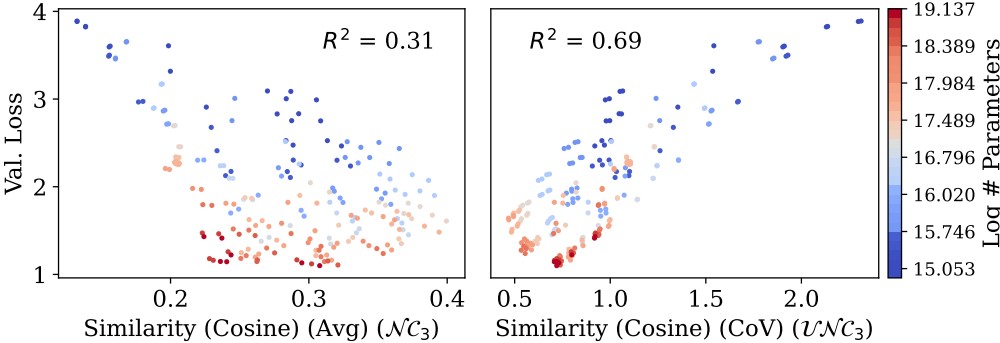

Figure 3: Validation loss shows a negligible relationship with self-duality ($\mathcal{NC}_3$, left) and some correlation with uniform duality ($\mathcal{UNC}_3$, right). In other words, $\mathcal{UNC}_3$ develops with scale and correlates with generalization much better than $\mathcal{NC}_3$.

### 4.4   Classifier Agreement — $\mathcal{NC}4$

The linear and NCC classifiers agree on far more examples than random chance, and model scaling encourages this agreement (Appendix Figs. 29, 30). Increasing width for certain depths happens to plateau or even regress the agreement rate, but this limitation is overcome with further training. And finally, agreement is a strong indicator of generalization (Fig. 1, right, $\mathcal{NC}_4$).

## 5   Analysis

We find that $\mathcal{NC}$ is generally promoted by model size and training and correlated with generalization (validation performance). We also discern some of this correlation independent of scale (§5.1).

Table 1: Permutation test of $\mathcal{NC}$ measurements with respect to validation loss. Twenty-one (21) identical two-layer 768-wide models were trained with different data shuffling seeds and permuted with $10^4$ trials. The $p$-values below 0.05 (bolded) show those properties to be statistically significant.

| Property | Measurement | $R^2$ Correlation ($\uparrow$) | $p$-value ($\downarrow$) |
|---|---|---|---|
| $\mathcal{NC}_1$ | Within-Class Variability Collapse | 0.192011 | **0.0485** |
| $\mathcal{NC}_2$ | Equinormness | 0.026174 | 0.4870 |
| $\mathcal{NC}_2$ | Equiangularity | 0.218574 | **0.0317** |
| $\mathcal{GNC}_2$ | Hyperspherical Uniformity | 0.487935 | **0.0002** |
| $\mathcal{NC}_3$ | Self-Duality | 0.322210 | **0.0063** |
| $\mathcal{UNC}_3$ | Uniform Duality | 0.000036 | 0.9784 |
| $\mathcal{NC}_4$ | Classifier Agreement | 0.490278 | **0.0001** |

## 5.1 Neural Collapse's Relationship with Generalization

Table 1 presents the correlation scores of $\mathcal{NC}$ metrics with generalization alongside their associated p-values from the permutation tests described in §3.8. The majority of the correlations are statistically significant ($p < 0.05$) independent of scale, affirming that $\mathcal{NC}$ is correlated with generalization.

## 5.2 Duality of Duality

The dichotomy of self-duality ($\mathcal{NC}_3$) and uniform duality ($\mathcal{UNC}_3$) is rather conspicuous. Our main experiments find that $\mathcal{NC}_3$ does not consistently develop with scale while $\mathcal{UNC}_3$ does (Fig. 3). However, within a fixed scale, the opposite is true, implying that $\mathcal{UNC}_3$ may be confounded by model capacity while $\mathcal{NC}_3$ is a subtle and fine-grained indicator of generalization.

## 5.3 The Effect of Weight Regularization

Our models trained with either weight decay factor exhibited very similar patterns in the emergence of $\mathcal{NC}$ (or lack thereof), but the more aggressive factor $\beta = 0.1$ resulted in stronger development of $\mathcal{NC}$ properties than with $\beta = 0.0005$ (Appendices E, F, G, H, I, J, K, M, N, P). These findings empirically affirm $\beta = 0.1$ weight decay as sensible for CLM pre-training [81], and concur with [51] on the pivotal role that appropriate regularization plays in the emergence of $\mathcal{NC}$.

# 6 Limitations

**Neural collapse**   While to the best of our knowledge, no previous work has studied realistic stochastic token prediction, it is possible that the quantities that we measure are not perfectly suited for $\mathcal{NC}$ in language modelling. As we described in §1.1, the $\mathcal{NC}$ framework does not translate neatly to the language modelling space due to many adverse conditions, so full convergence to $\mathcal{NC}$ in the TPT was highly improbable. This paper leaves much room for future work to better adapt $\mathcal{NC}$ for next-token prediction, which we discuss further in Section 7.

**Language modelling**   Our work focused on autoregressive pre-training in its most basic form. We did not conduct experiments into encoder, multi-modal, or instruction-tuned models. Post-training techniques such as supervised fine-tuning, reinforcement learning with human feedback [91] or direct preference optimization [92] are also out-of-scope. This paper uses validation CE loss as the sole indicator of performance, leaving out any downstream task evaluations.

**Confounder of model scale**   The models that we used in our permutation test (§5.1, Table 1) are only of a single small architecture trained for one epoch with relatively weak weight regularization ($\beta = 0.0005$). Therefore, our experimental results on scale-independent links between $\mathcal{NC}$ and generalization may not necessarily translate to larger models. Further investigation on (foundation) LLMs orders of magnitude larger than our CLMs trained with modern NLP methods would provide more robust insight into any direct correlations.

# 7 Discussion

**Layer/depth-wise neural collapse**   Past works have established that properties resembling $\mathcal{NC}$ evolve as a function of model depth [4, 36, 53, 56, 59, 60, 93–104]. Layer-wise $\mathcal{NC}$ — sometimes dubbed *deep neural collapse* ($\mathcal{DNC}$) [53, 59] — and related phenomena at intermediate layers remain an interesting subtopic. We leave their study and induction in CLMs (like [105]) as future work.

**Learning to collapse**   Given the evidence for the development of $\mathcal{NC}$ and associated generalization under various loss functions [19, 49, 60, 100, 106–108] in other domains, NLP researchers may still benefit from analyzing, adapting or training towards $\mathcal{NC}$. As alluded to earlier, the simplex ETF and even the CE loss may not be truly optimal for this problem setting, so we anticipate future works to both construct more amenable geometries with better-suited objectives and then capitalize on their benefits downstream. As discussed in §2, there is an abundance of literature in $\mathcal{NC}$, some of which could potentially adapt $\mathcal{NC}$ to be useful for NLP; we hope to inspire more such investigations.

**Interpretability**   At a high level, the number and density of clusters for a token can reflect its learned meanings and uses. This would be particularly useful as LLMs adapt to ever-evolving language and further expansion into non-English domains. Our formulae (Section 3) and results (Section 4) expose the pairwise token class interactions in noise, interference, classifier duality, and classifier agreement in the top-level features. Similarly to works in other domains [21, 55, 72, 109], these $\mathcal{NC}$ metrics can serve as a form of low-level interpretability to aid understanding certain behaviours of (L)LMs. Between tokens, one can often discern how related or interchangeable words are based on their pair-wise interactions, or how antithetical or unrelated they are based on orthogonality. For example, we present a rudimentary analysis of homonyms and English first names in Appendix Q.

**Fairness**   Foundation models are ubiquitous for their comprehensive capabilities and adaptability. As previous work discussed class imbalance [35, 58, 61, 62], our work may extend these strategies to measure and perhaps promote fairness in foundation LLMs, some of which are designed to be multilingual or multicultural. For example, [110] contemporarily explores the use of $\mathcal{UNC}3$ to mitigate biases in BERT-based [7] models.

While $\mathcal{NC}$ itself would not lead to unfairness, its potential interpretability may, in theory, enable an (adversarial) agent to measure and optimize for (un)fairness as they manipulate an LLM.

**LLM Evaluations**   Researchers in NLP and multimodal settings are ultimately interested in measuring model performance on practical tasks; notable benchmarks include GLUE [77], MMLU [111], and BIG-bench [112]. However, several contemporaries [113–115] have demonstrated that models' downstream capabilities are roughly correlated with their abilities to effectively compress their pre-training data. Based on their findings, our application of the $\mathcal{NC}$ framework to the pre-training stage of CLMs against validation CE loss should be an appropriate first step in this intersection. Looking forward, we anticipate exciting analysis for language modelling tasks or benchmarks, especially on creativity and retrieval for natural language understanding and generation.

Conversely, some form of $\mathcal{NC}$ could be an alternative evaluation. Although it would be prohibitively expensive to measure $\mathcal{NC}$ on the vast and sometimes obscure pre-training data of most frontier production LLMs, doing so on a small set of in-distribution data (i.e. test set) would be realistic.

# 8 Conclusion

In this paper, we apply the *neural collapse* ($\mathcal{NC}$) framework to the next-token prediction problem, where models are undertrained and next-tokens are variably drawn from numerous and imbalanced token classes. We leverage canonical and more recent metrics to demonstrate that $\mathcal{NC}$ emerges as we scale the size and training of hundreds of causal language models. Our results show a correlation between $\mathcal{NC}$ and generalization, a relationship that persists even when the model scale is fixed.

In the short term, our work presents rudimentary techniques to analyze and interpret token-level properties of (L)LMs. We anticipate future work to suitably adapt $\mathcal{NC}$ (and related frameworks) to the still-fresh frontier of autoregressive language modelling. Researchers could then effectively capitalize on previous learnings from $\mathcal{NC}$ to better understand and improve the pre/post-training processes of increasingly complex and large language (multimodal) models.

## Acknowledgements

We thank Elliot Creager, David Glukhov, Daniel Johnson, Jivan Waber, and Colin Raffel for their helpful feedback and stimulating discussions. Aditya Mehrotra and Yu Bo Gao provided technical assistance in our implementations. We acknowledge the support of the Natural Sciences and Engineering Research Council (NSERC) of Canada (`www.nserc-crsng.gc.ca/`). This research was enabled in part by resources from Calcul Québec (`www.calculquebec.ca`), the Digital Research Alliance of Canada (`www.alliancecan.ca`), and the Vector Institute (`www.vectorinstitute.ai`).

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

# A Dataset

TinyStories is a dataset of short children's stories generated by GPT-3.5 and GPT-4 [2], released with the CDLA-Sharing-1.0 licence. We trained and evaluated models on their first version, as described:

- The 2,141,709 stories are split into 2,119,719 train and 21,990 validation stories.
- Their experimental setup [2] called for the GPT-2 [68] tokenizer, of which only a subset vocabulary $\mathbb{V} = [\![1, 29233]\!]$ appears in TinyStories.
- Following the GPT-style teacher-forcing regime for training/evaluation [8], raw sequences (stories) from the train set are packed (by two preprocessing workers) into 229,367 ($S$) chunks of 2048 ($T$) tokens each. This setup provides 469,514,249 ($N$) ground-truth[17] token samples for training.

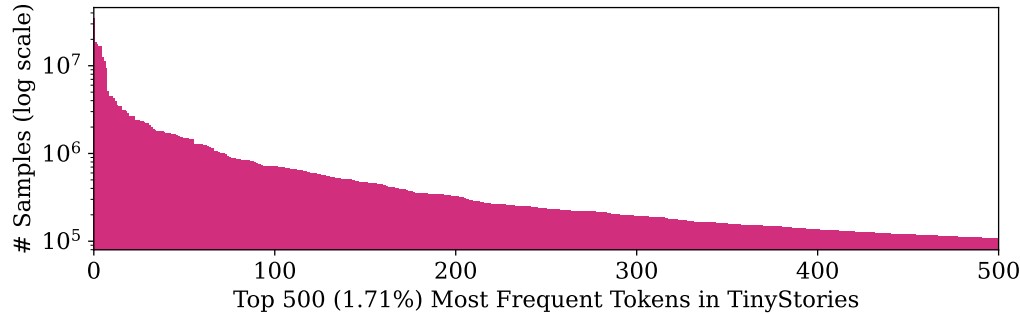

Figure 4: The 500 most frequent classes from TinyStories [2] exhibit significant sample imbalance. Despite the synthetic nature of TinyStories, such a distribution is typical of natural language [11, 12].

## A.1 Alternative (Real) Datasets

The study of $\mathcal{NC}$ in causal language modelling at the token level would be unreasonably expensive, so the motivation to use a small dataset is clear. However, most commonly used text datasets such as WikiText [116], BookCorpus [117], CommonCrawl [118], or most subsets from the Pile [119] are much too complex and broad to be effectively compressed by CLMs of the scale that we work with.

WikiText-2 and WikiText-103 present significant drawbacks for our experiments. Both datasets contain a considerable amount of low-quality data that does not concentrate on essential linguistic structures such as grammar, vocabulary, facts, and reasoning. WikiText-2 has a similar empirical vocabulary to TinyStories under the GPT-Neo [80] tokenizer (27K vs. 29K) but only has around 34K rows of training data compared to 2.1M in TinyStories. Our small-scale NC experiment on WikiText-2 revealed that the models were very brittle and prone to overfitting. On the other hand, WikiText-103 is comparably sized to TinyStories but utilizes around 44K unique tokens. Our CLMs trained on WikiText-103 struggled to produce coherent sentences, likely due to the excessive breadth and information, as noted by the authors of TinyStories. Beyond these two, we were unable to find any real datasets that both followed established scaling laws [14, 15] for CLMs at our scale and are simple enough to suit the analysis of $\mathcal{NC}$.

## A.2 On the Use of TinyStories

According to its authors, TinyStories [2] is explicitly designed to preserve the essential elements of natural language, such as grammar, vocabulary, facts, and reasoning, while being smaller and more refined in terms of its breadth and diversity. Unlike large corpora that can overwhelm small language models (SLMs) due to their excessive breadth and diversity, TinyStories offers a concentrated dataset that hones in on core linguistic structures and reasoning capabilities. This is evident in its small vocabulary, consisting of approximately 1500 words that a child would use, and in its 29K empirical vocabulary under the GPT-Neo tokenizer.

---

[17]$N = S(T - 1)$ as we cannot use the first ground-truth nor the last predicted token in any chunk.

Despite its concentrated nature, TinyStories enables models trained on it to produce grammatically correct, factual, and reasonable stories. Additionally, these models can be finetuned on specific instructions found in the TinyStories-Instruct dataset. The authors of TinyStories also demonstrate that their models can creatively produce stories dissimilar enough to their training data, indicating a balanced capability for generalization and creativity.

One particular advantage of TinyStories is the small vocabulary relative to total training tokens, rendering a reasonable number of classes with higher average token counts. This is relevant because the possibility of $\mathcal{NC}$ and a CLM's ability to compress language data into distinct geometries depend partially on the ratios between embedding dimension, vocabulary size, and average token frequency. Conveniently, frequency analysis of the overall dataset produced a distribution (Figure 4) similar to real human language, so TinyStories should provide a good balance for an initial study of this $\mathcal{NC}$.

Additionally, TinyStories has a more regular structure as GPT-3.5/4 was instructed to produce children's stories with certain themes and forms with a conservative vocabulary. We believe this would reduce the amount of clustering noise from the breadth of information and structures in real general data, and allow our smaller CLMs to exhibit some clear trends toward $\mathcal{NC}$.

Furthermore, TinyStories was created using GPT 3.5/4, advanced language models with significantly larger architectures trained on orders of magnitude more tokens; this should help minimize the effect of the synthetic nature of the generated dataset. We also considered a possible effect of model collapse as a result of training on synthetic data [120] and follow-up work [121] suggest that a single iteration of data generation (as generated TinyStories) has a very negligible model collapse.

## B  Model Architectural Details

Table 2: Sample architectural configuration for a 12-layer 1024-dimensional causal language model (CLM) based on [2] and GPT-Neo [80]. Shallower models have configurations with `attention_layers` and `attention_types` truncated. Narrower models adjust `hidden_size` to their width ($d$). All other configuration values are the same across models.

| SETTING | VALUE |
|---|---|
| activation_function | gelu_new |
| architectures | GPTNeoForCausalLM |
| attention_dropout | 0 |
| attention_layers | global, local, global, local, ... |
| attention_types | [[global, local], 6] |
| bos_token_id | 50256 |
| embed_dropout | 0 |
| eos_token_id | 50256 |
| gradient_checkpointing | false |
| hidden_size | 1024 |
| initializer_range | 0.02 |
| intermediate_size | null |
| layer_norm_epsilon | 1e-05 |
| max_position_embeddings | 2048 |
| model_type | gpt_neo |
| num_heads | 16 |
| num_layers | 12 |
| resid_dropout | 0 |
| summary_activation | null |
| summary_first_dropout | 0.1 |
| summary_proj_to_labels | true |
| summary_type | cls_index |
| summary_use_proj | true |
| torch_dtype | float32 |
| transformers_version | 4.28.1 |
| use_cache | true |
| vocab_size | 50257 |
| window_size | 256 |

## C   Optimization

The training was performed using an adaptation of an open-source causal language modelling script from Huggingface: `https://github.com/huggingface/transformers/blob/main/examples/pytorch/language-modeling/run_clm.py`

- Each model was trained on a single NVIDIA A100 (40GB) GPU for up to 8 hours per epoch.
- Learning rates were set by a linear schedule based on the number of steps with no warm-up.
- Training was performed in `bfloat16` [122] mixed precision.
- The results presented in this work are from two sets of models trained with weight decay $\beta = 0.0005$ [51] and $\beta = 0.1$ [81]. A previous set of models was trained without weight decay and the results are very similar to $\beta = 0.0005$.

Table 3: Batch sizes used to train models on a single NVIDIA A100 (40GB) GPU. Width ($d$) and depth ($L$) correspond to `hidden_size` and length of `attention_layers`, respectively, in Table 2.

| **DEPTH** ($L$)↓   **WIDTH** ($d$) → | 64 | 128 | 256 | 512 | 768 | 1024 |
|---|---|---|---|---|---|---|
| 1-layer | 16 | 16 | 16 | 16 | 16 | 16 |
| 2-layer | 16 | 16 | 16 | 16 | 16 | 8 |
| 4-layer | 8 | 8 | 8 | 8 | 8 | 8 |
| 8-layer | 8 | 8 | 8 | 4 | 4 | 4 |
| 12-layer | 4 | 4 | 4 | 4 | 4 | 4 |

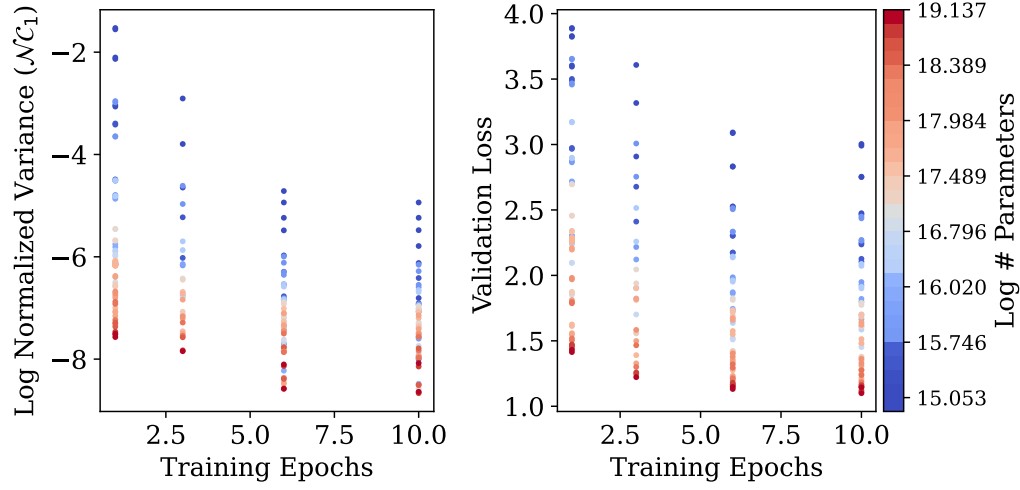

Figure 5: Average (logarithmic) class-distance normalized variance (CDNV, $\mathcal{NC}_1$) (left) and validation (cross-entropy) loss (right) with respect to training epochs.

## D   Embeddings Collection & $\mathcal{NC}$ Analysis

Codes for (post-)training analysis are hosted on GitHub:

- Main code `https://github.com/rhubarbwu/linguistic-collapse`
- Auxillary package: `https://github.com/rhubarbwu/neural-collapse`

One pass over the train set for embeddings collection can take up to 6 hours on a single NVIDIA A100 (40GB) GPU. Analysis of a single metric for a given model takes less than 5 minutes.

# E    Within-Class Variability Collapse with Scale — $\mathcal{NC}_1$

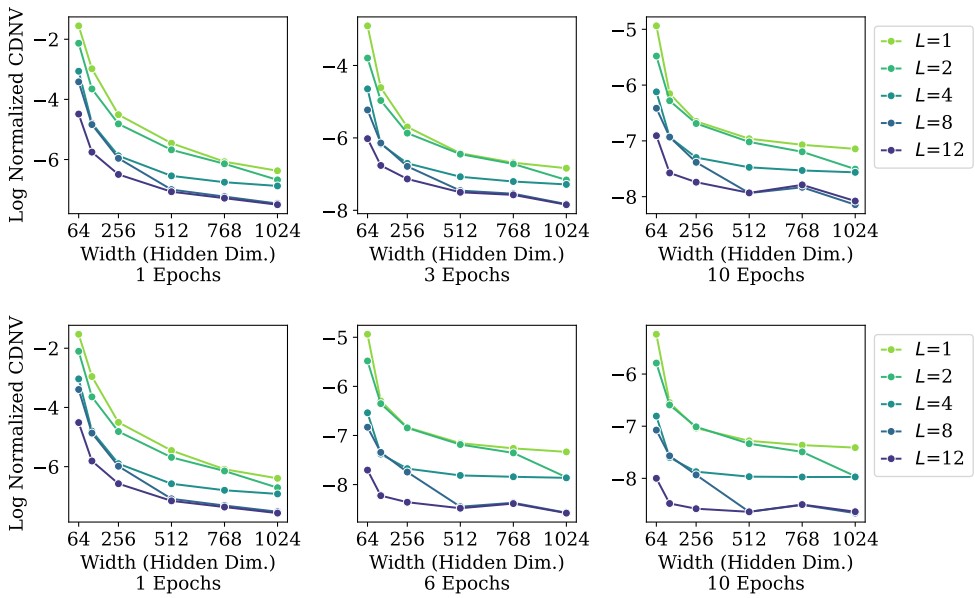

Figure 6: Average (logarithmic) class-distance normalized variance (CDNV) is reduced ($\mathcal{NC}_1$) when scaling width ($d$) and across training for 1 (left) through 10 (right) epochs with weight decays $\beta = 0.0005$ (top) and $\beta = 0.1$ (bottom).

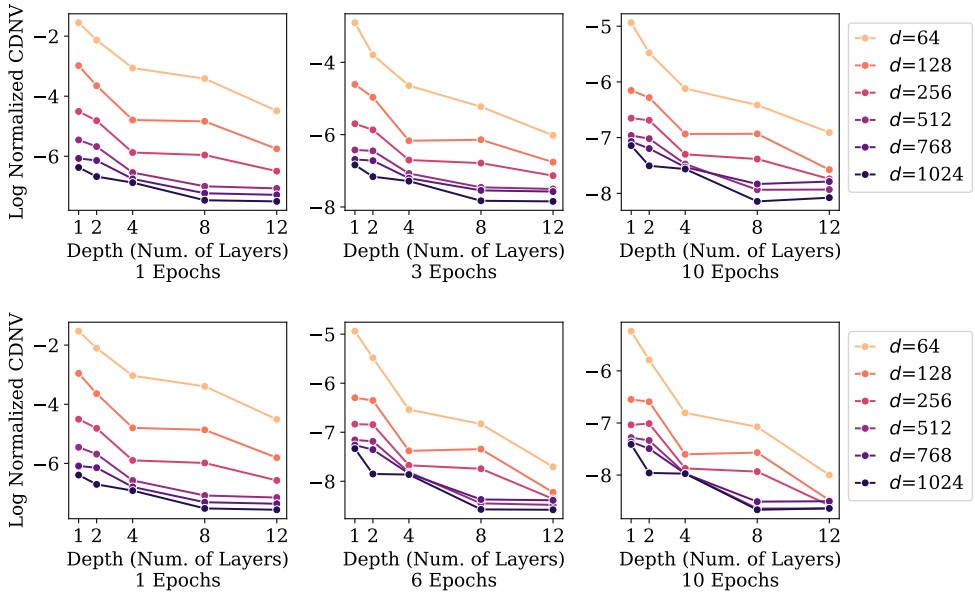

Figure 7: Average (logarithmic) class-distance normalized variance (CDNV) is reduced ($\mathcal{NC}_1$) when scaling depth ($L$) and across training for 1 (left) through 10 (right) epochs with weight decays $\beta = 0.0005$ (top) and $\beta = 0.1$ (bottom).

## F  Mean Norms Growth with Scale — (Related to $\mathcal{NC}_2$)

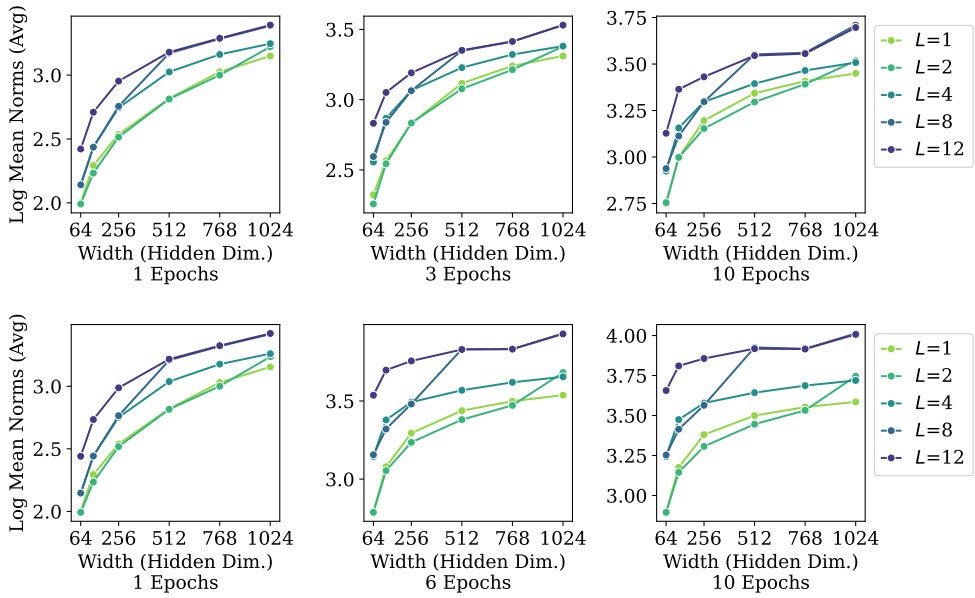

Figure 8: Logarithmic class mean norms grow when scaling width ($d$) and across training for 1 (left) through 10 (right) epochs with weight decays $\beta = 0.0005$ (top) and $\beta = 0.1$ (bottom).

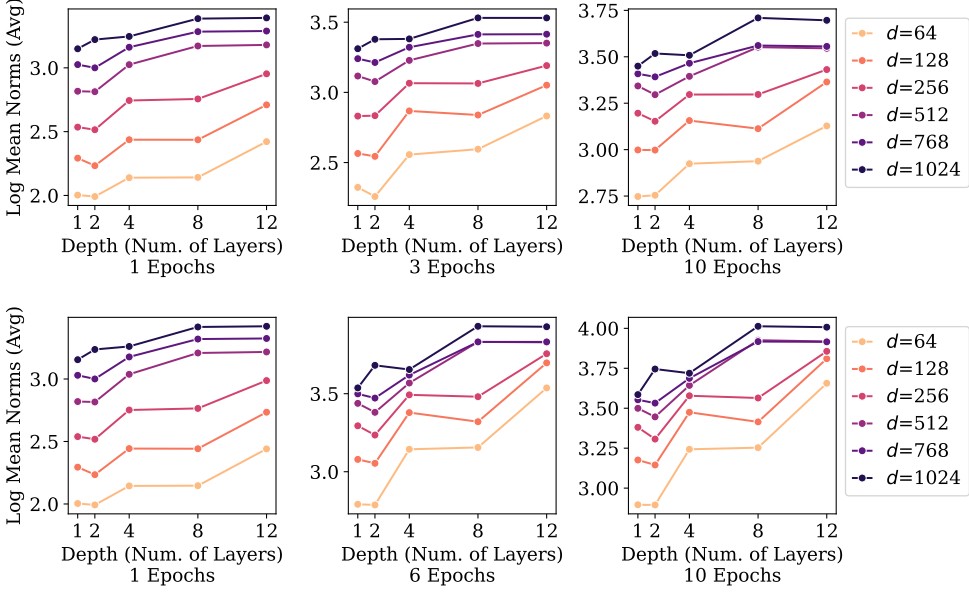

Figure 9: Logarithmic class mean norms grow when scaling depth ($L$) and across training for 1 (left) through 10 (right) epochs with weight decays $\beta = 0.0005$ (top) and $\beta = 0.1$ (bottom).

# G   Equinormness with Scale — $(\mathcal{G})\mathcal{N}\mathcal{C}_2$

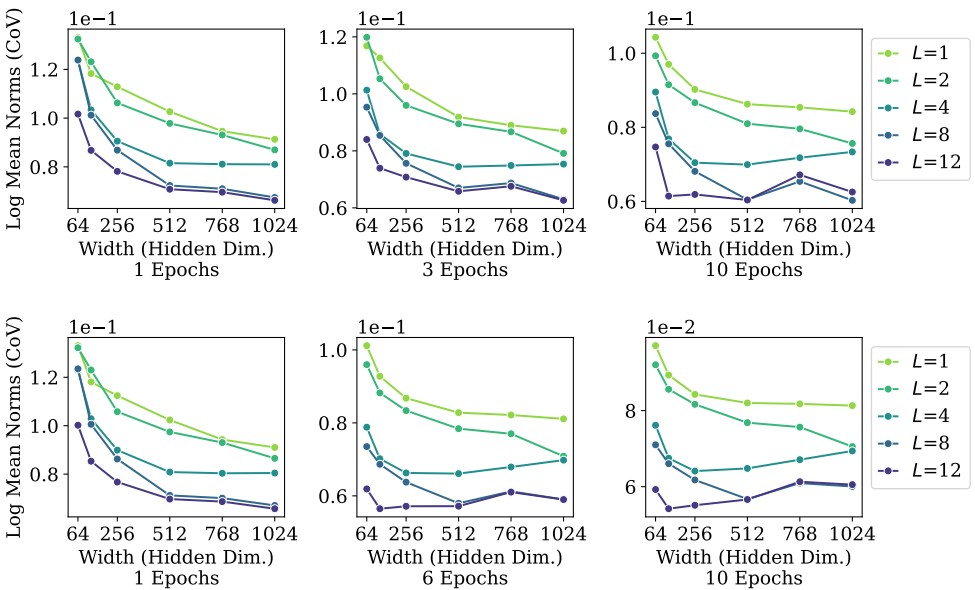

Figure 10: Variation in (logarithmic) norms decreases ($\mathcal{N}\mathcal{C}_2$) when scaling width ($d$) in models trained for 1 (left) through 10 (right) with weight decays $\beta = 0.0005$ (top) and $\beta = 0.1$ (bottom). Note that the degree of equinormness eventually plateaus for sufficiently deep and trained models.

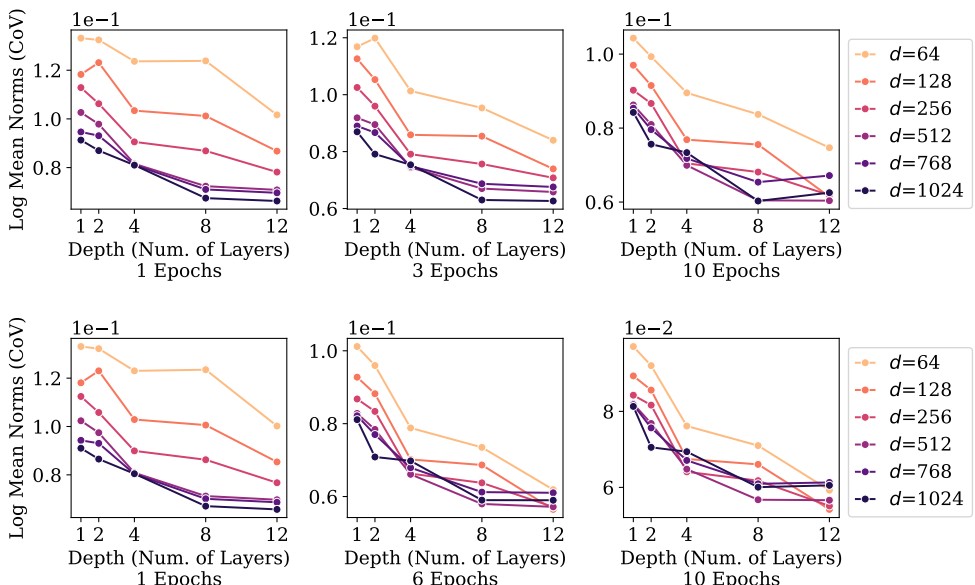

Figure 11: Variation in (logarithmic) norms decreases ($\mathcal{N}\mathcal{C}_2$) when scaling depth ($L$) in models trained for 1 (left) through 10 (right) with weight decays $\beta = 0.0005$ (top) and $\beta = 0.1$ (bottom).

## H  Interference with Scale — (Related to $\mathcal{NC}_2$)

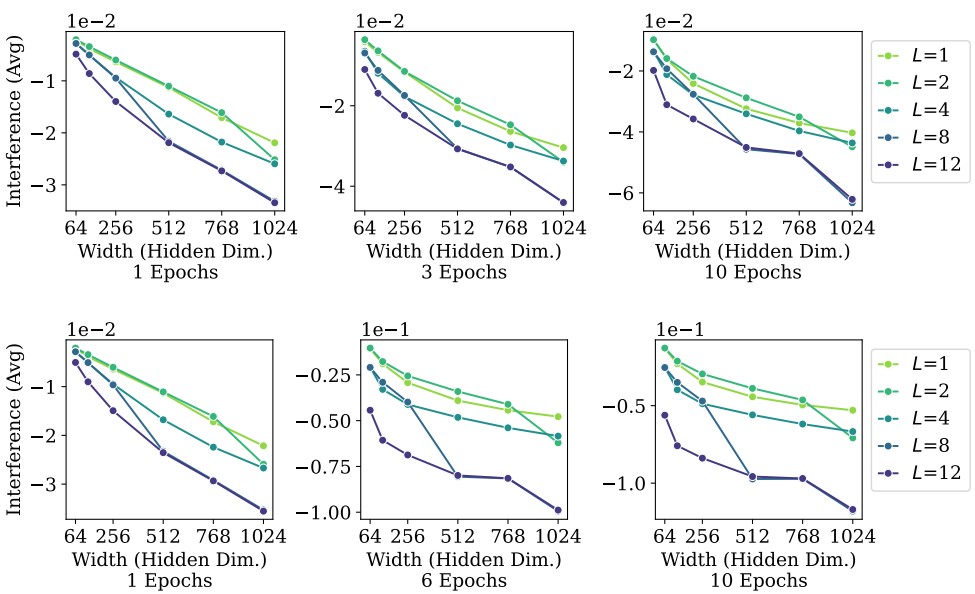

Figure 12: Average interference decreases (to some extent) when scaling width ($d$) in models trained for 1 (left) through 10 (right) with weight decays $\beta = 0.0005$ (top) and $\beta = 0.1$ (bottom).

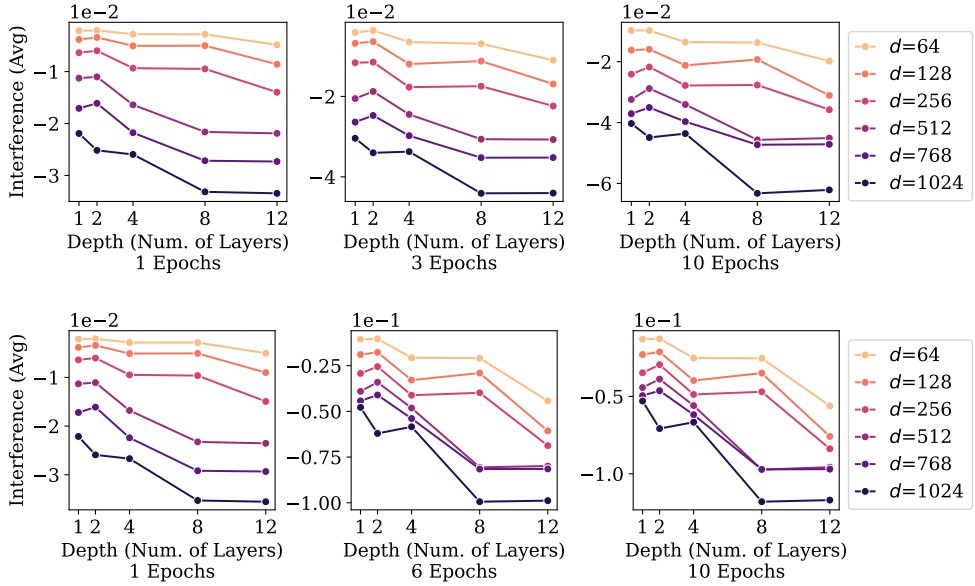

Figure 13: Average interference decreases (to some extent) when scaling depth ($L$) in models trained for 1 (left) through 10 (right) with weight decays $\beta = 0.0005$ (top) and $\beta = 0.1$ (bottom).

# I Equiangularity with Scale — (Against $\mathcal{NC}_2$)

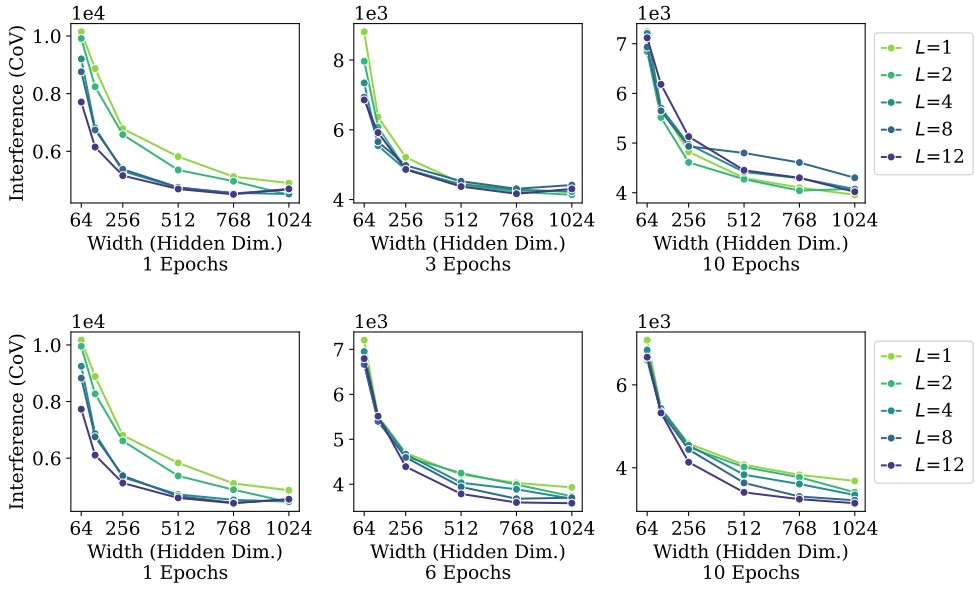

Figure 14: Variation in interference roughly increases when scaling width ($d$) in models trained for 1 (left) through 10 (right) with weight decays $\beta = 0.0005$ (top) and $\beta = 0.1$ (bottom). Note this trend is against equiangularity, affirming the traditional $\mathcal{NC}2$ to be less useful than $\mathcal{GNC}2$ [19].

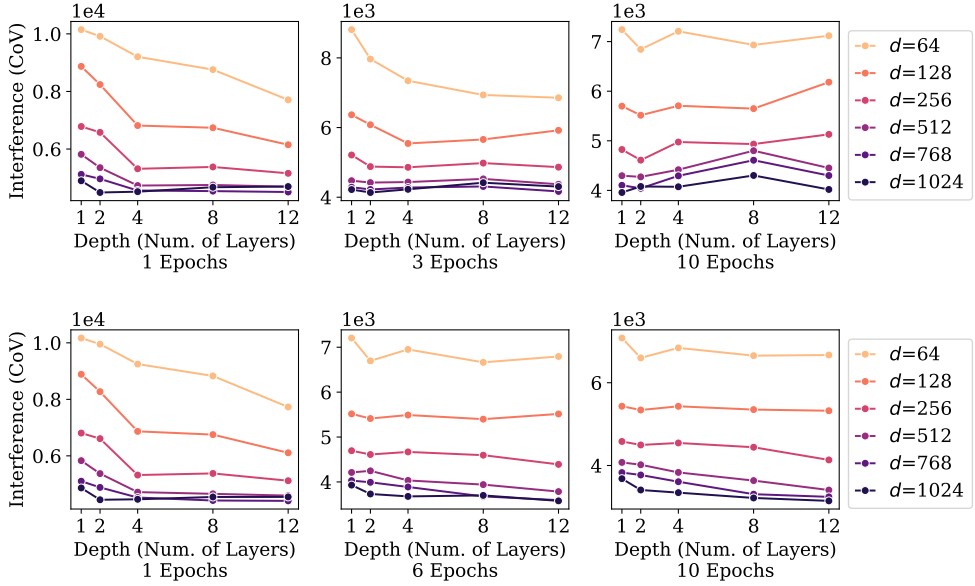

Figure 15: Variation in interference increases when scaling depth ($L$) in models trained for 1 (left) through 10 (right) with weight decays $\beta = 0.0005$ (top) and $\beta = 0.1$ (bottom). Note this trend is against equiangularity, affirming the traditional $\mathcal{NC}_2$ to be less useful than $\mathcal{GNC}_2$ [19].

## J  Logarithmic Distances with Scale — (Related to $\mathcal{GNC}_2$)

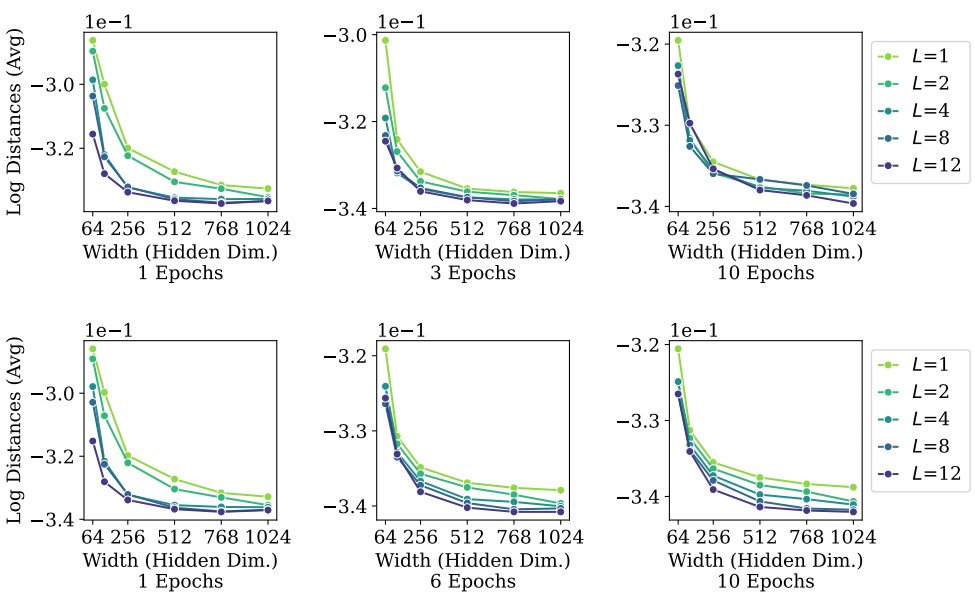

Figure 16: Average logarithmic distance decreases when scaling width ($d$) in models trained for 1 (left) through 10 (right) with weight decays $\beta = 0.0005$ (top) and $\beta = 0.1$ (bottom).

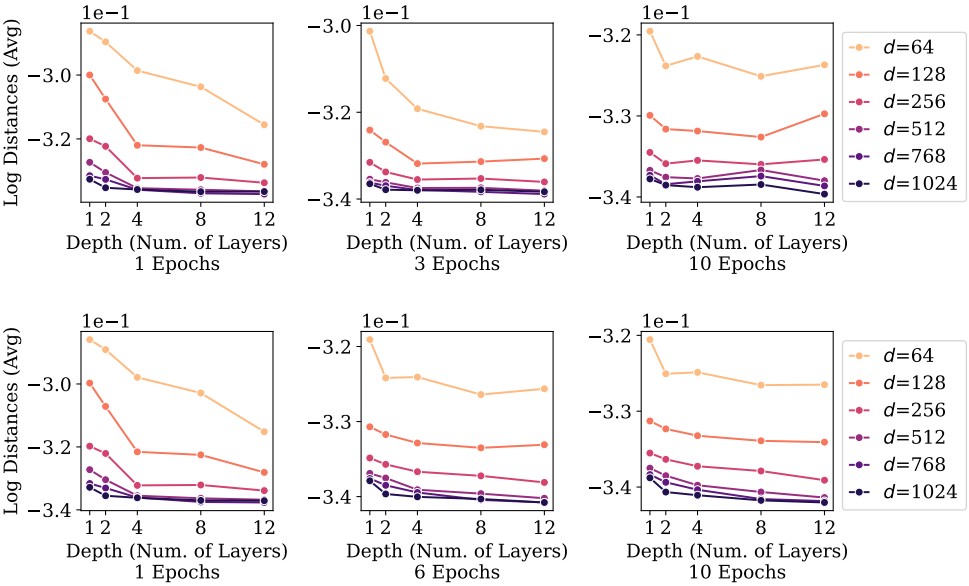

Figure 17: Average logarithmic distance decreases when scaling depth ($L$) in models trained for 1 (left) through 10 (right) with weight decays $\beta = 0.0005$ (top) and $\beta = 0.1$ (bottom).

# K    Hyperspherical Uniformity with Scale — $\mathcal{GNC}_2$

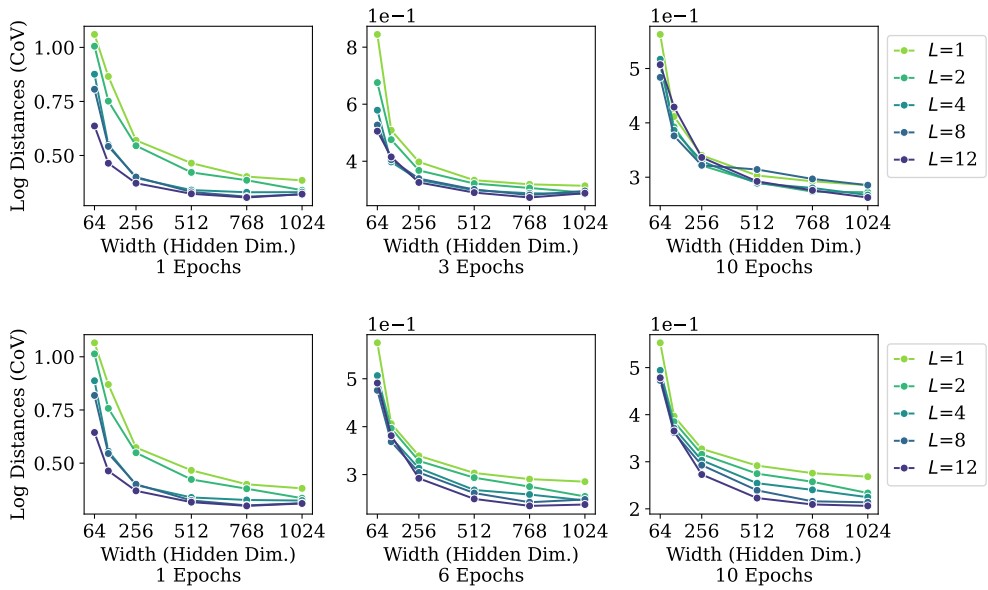

Figure 18: Variation in logarithmic distances decreases when scaling width ($d$) in models trained for 1 (left) through 10 (right) with weight decays $\beta = 0.0005$ (top) and $\beta = 0.1$ (bottom). This consistent trend towards hyperspherical uniformity affirms that $\mathcal{GNC}2$ [19] is more useful than $\mathcal{NC}2$.

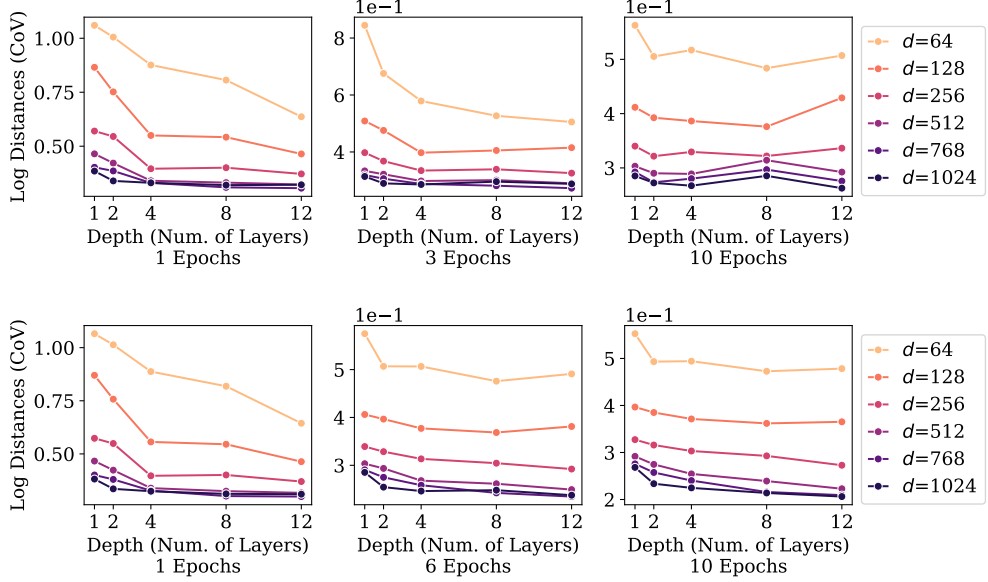

Figure 19: Variation in logarithmic distances decreases when scaling depth ($L$) in models trained for 1 (left) through 10 (right) with weight decays $\beta = 0.0005$ (top) and $\beta = 0.1$ (bottom). This consistent trend towards hyperspherical uniformity affirms that $\mathcal{GNC}2$ [19] is more useful than $\mathcal{NC}2$.

# L   Correlations of $(\mathcal{G})\mathcal{N}\mathcal{C}_2$ with Generalization Performance

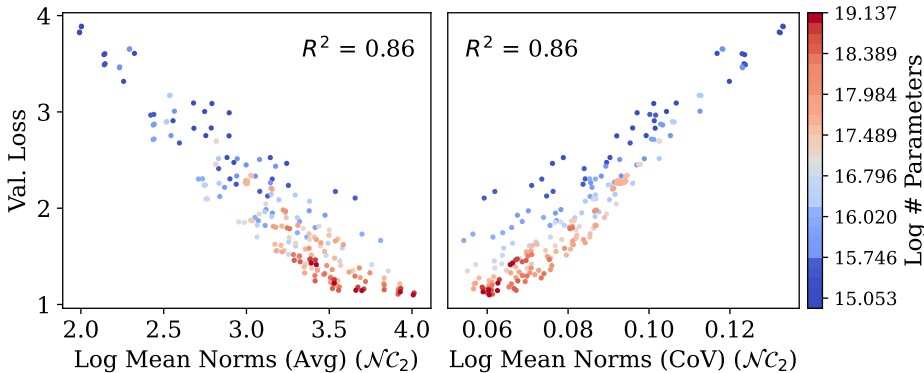

Figure 20: Generalization (validation loss) shows some correlation with logarithmic mean norms in both their average (left) and variations (i.e. equinormness, $\mathcal{N}\mathcal{C}_2$) (right).

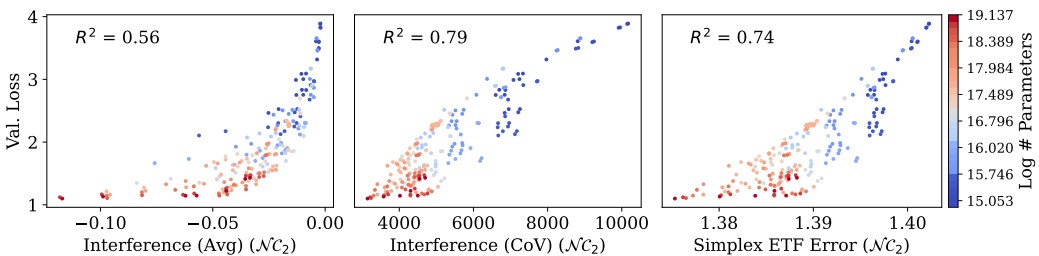

Figure 21: Generalization (validation loss) correlated with average interference (left) and its variation (i.e. equiangularity, $\mathcal{N}\mathcal{C}_2$) (centre). We also computed the empirical measure from [5] (right).

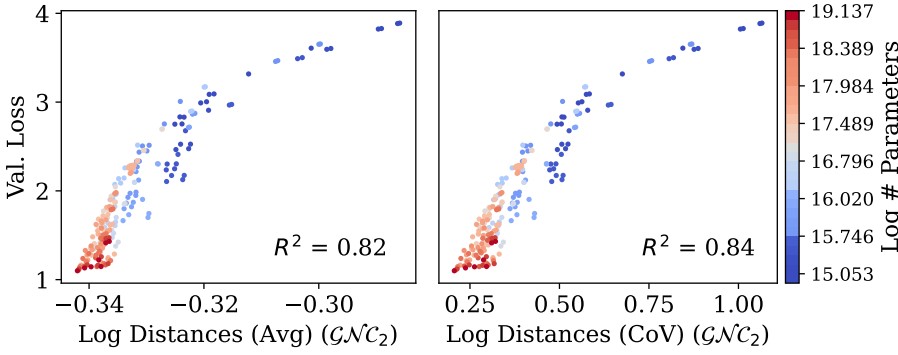

Figure 22: Validation loss shows some correlation with average (logarithmic) kernel distances and with their variation (i.e. hyperspherical uniformity, $\mathcal{G}\mathcal{N}\mathcal{C}_2$) (right).

# M Self-Duality with Scale — (Against $\mathcal{NC}_3$)

Self-duality ($\mathcal{NC}_3$) was originally the convergence of classifiers to means up to rescaling [1]:

$$\left\| \frac{\boldsymbol{w}_c}{\|\boldsymbol{w}_c\|_2} - \hat{\boldsymbol{\mu}}_c \right\|_2 \to 0, \quad \forall c \tag{11}$$

Instead, we use class-wise cosine similarity (Equation 8) and its variation ($\mathcal{UNC}3$).

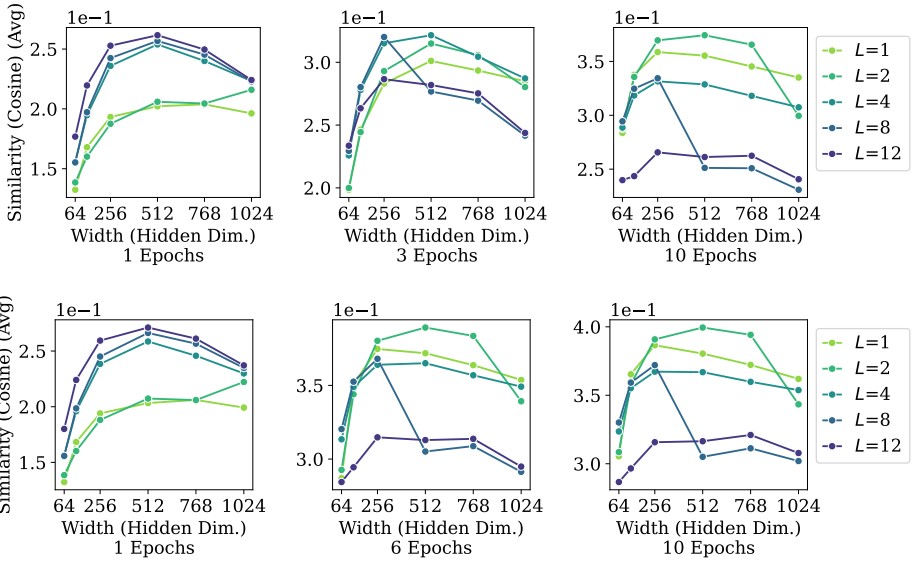

Figure 23: Average classifier alignment increases $\mathcal{NC}3$ when training for 1 (left) through 10 (right) with weight decays $\beta = 0.0005$ (top) and $\beta = 0.1$ (bottom). However, we see no meaningful trend when scaling width $d$, suggesting that $\mathcal{NC}3$ does not coalesce with language modelling training.

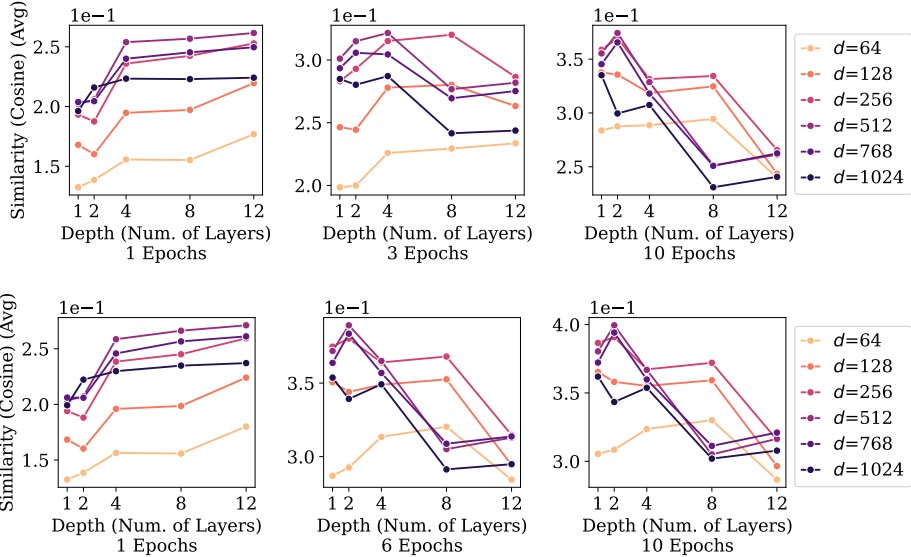

Figure 24: Average classifier alignment increases $\mathcal{NC}3$ when training for 1 (left) through 10 (right) with weight decays $\beta = 0.0005$ (top) and $\beta = 0.1$ (bottom). However, we see no meaningful trend when scaling depth $L$, suggesting that $\mathcal{NC}3$ does not coalesce with language modelling training.

# N   Uniformity Duality with Scale — $\mathcal{UNC}_3$

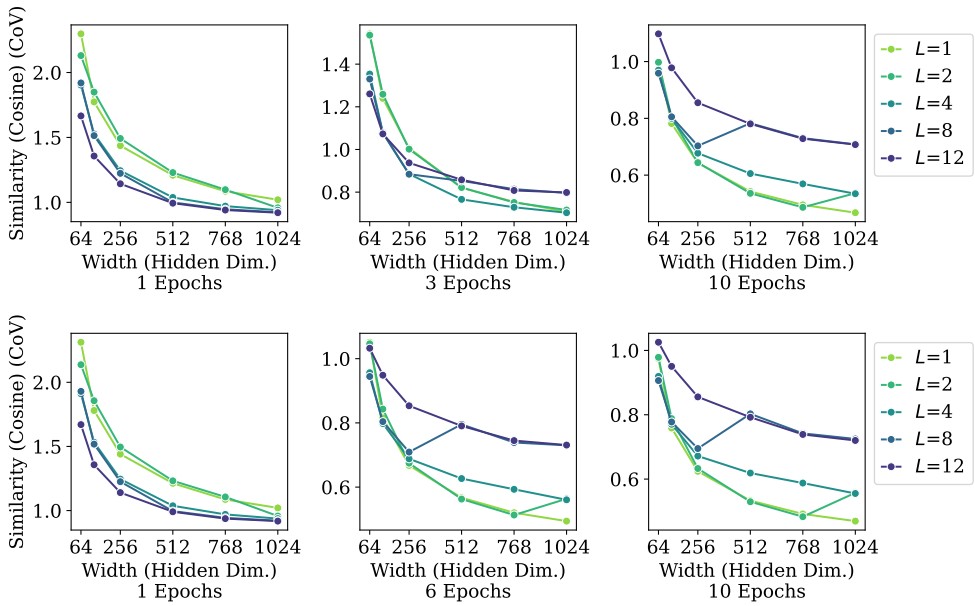

Figure 25: Variation in classifier alignment decreases when scaling width ($d$) in models trained for 1 (left) through 10 (right) with weight decays $\beta = 0.0005$ (top) and $\beta = 0.1$ (bottom).

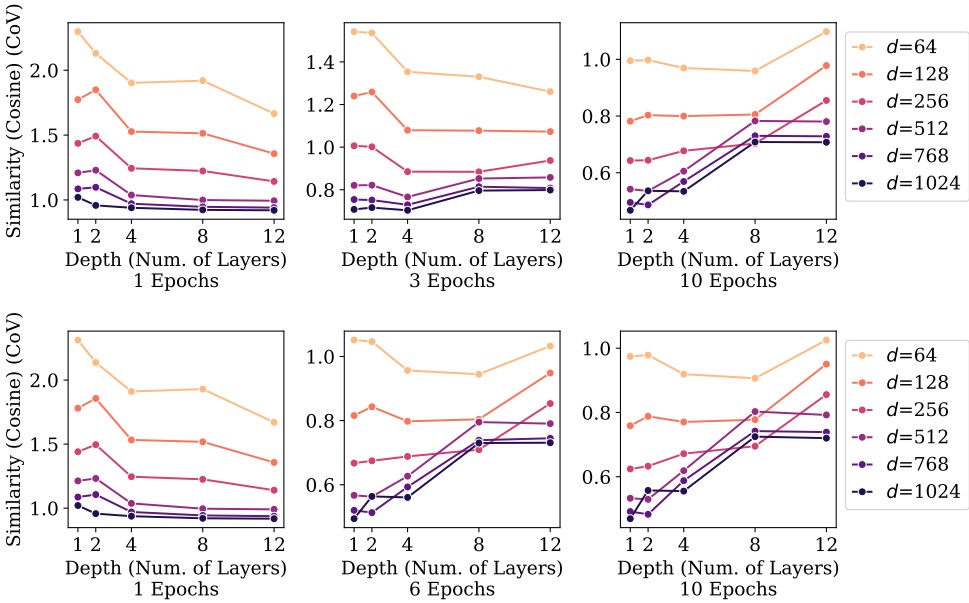

Figure 26: Variation in classifier alignment increases when scaling depth ($L$) in models trained for 1 (left) through 10 (right) with weight decays $\beta = 0.0005$ (top) and $\beta = 0.1$ (bottom). This negative trend of $\mathcal{UNC}3$ in more learnt models (right) suggests that the link of $(\mathcal{U})\mathcal{NC}3$ with scale and performance is still weak.

# O Correlations of $(\mathcal{U})\mathcal{NC}_3$ with Generalization Performance

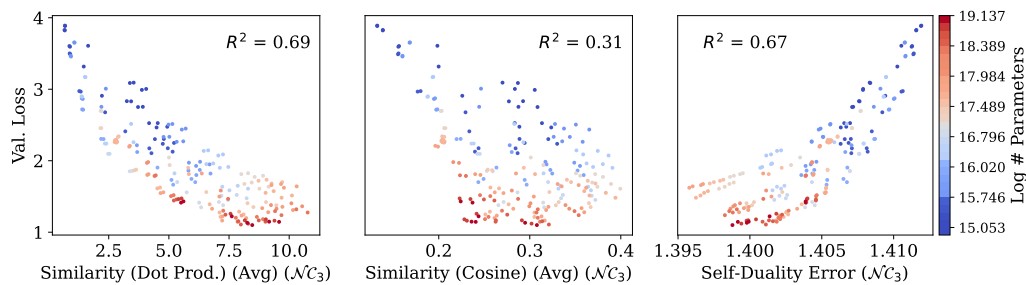

Figure 27: Generalization (validation loss) correlated with average dot-product similarity (for interpretability purposes only) (left) and cosine similarity (classifier alignment, $\mathcal{NC}_3$) (centre). We also computed the empirical measure from [5] (right).

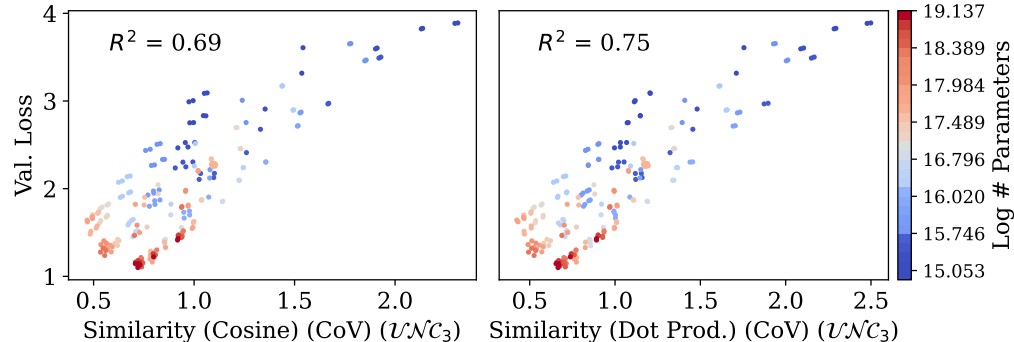

Figure 28: Generalization (validation loss) correlated with variation in dot-product similarity (for interpretability purposes only) (left) and cosine similarity (uniform duality, $\mathcal{UNC}_3$) (right).

# P  Classifier Agreement — $\mathcal{NC}_4$

For computational reasons, we compute Equations 9, 10 using a simple decomposition:

$$\underset{c\in\mathbb{V}}{\operatorname{argmin}}\,\|\boldsymbol{h}_b-\boldsymbol{\mu}_c\|_2 = \underset{c\in\mathbb{V}}{\operatorname{argmin}}\left(\|\boldsymbol{h}_b\|^2+\|\boldsymbol{\mu}_c\|^2-2\boldsymbol{h}_b^\top\boldsymbol{\mu}_c\right), \tag{12}$$

where $b\in[1,B]$ and $c\in\mathbb{V}$ with batch size $B$ and vocabulary $\mathbb{V}$.

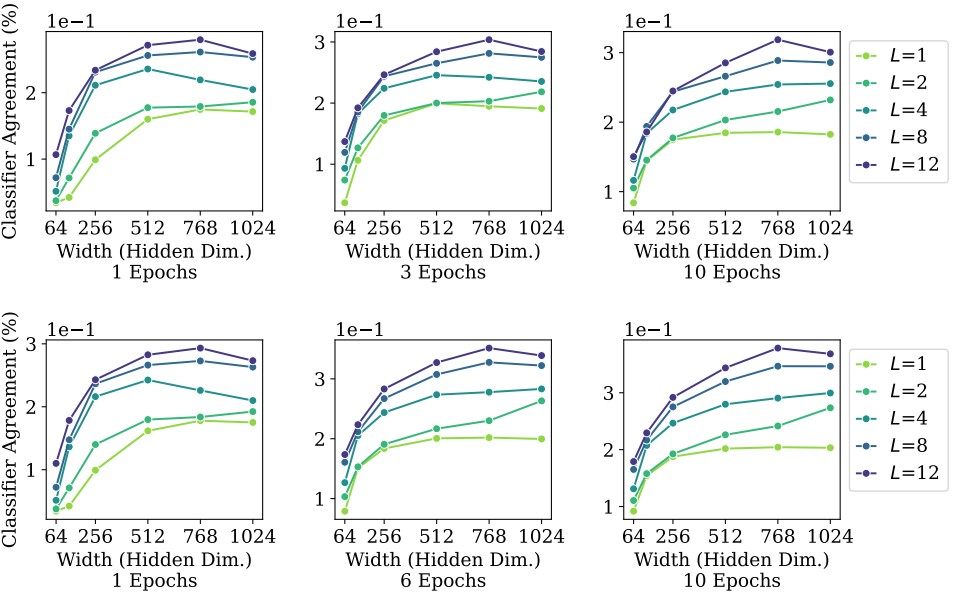

Figure 29: Classifier agreement improves when scaling width ($d$) in models trained for 1 (left) through 10 (right) with weight decays $\beta=0.0005$ (top) and $\beta=0.1$ (bottom).

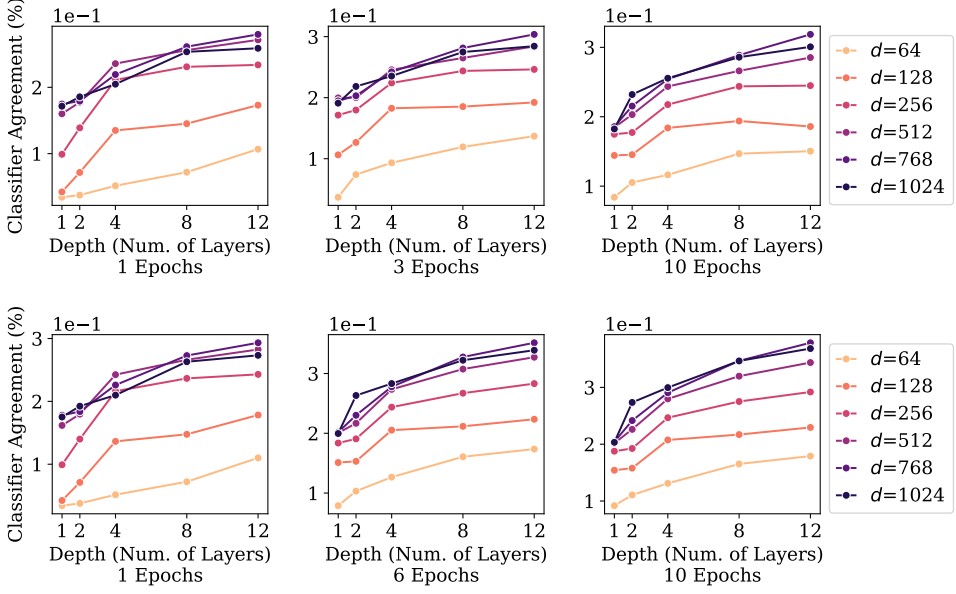

Figure 30: Classifier agreement improves when scaling depth ($L$) in models trained for 1 (left) through 10 (right) with weight decays $\beta=0.0005$ (top) and $\beta=0.1$ (bottom).

# Q    Examples for Interpretability

This section presents token-wise interpretability results from top-layer embeddings from our most learned models (trained for 10 epochs). Our largest one is publicly available: `https://huggingface.co/rhubarbwu/TinyStories-12x1024_10L`.

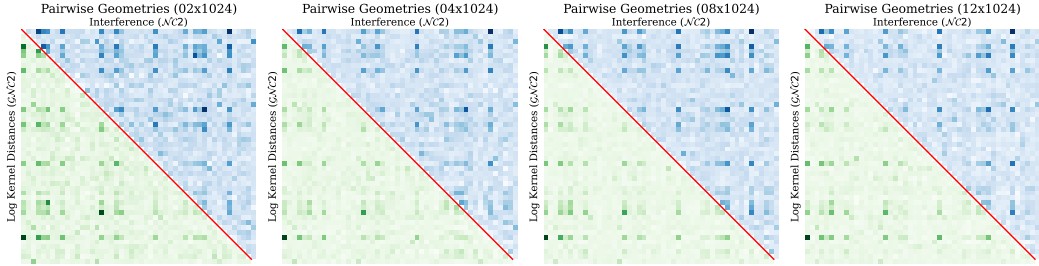

Figure 31: Pairwise interactions between token mean embedding vectors across models of fixed width ($d = 1024$) and increasing depths ($L = 2, 4, 8, 12$). Interference ($\mathcal{NC}2$) decreases on average but only slightly becomes more uniform (top-right, blue). In contrast, logarithmic kernel distances ($\mathcal{GNC}2$) decrease and become more evenly spread, with some outlier pairs (bottom-left, green).

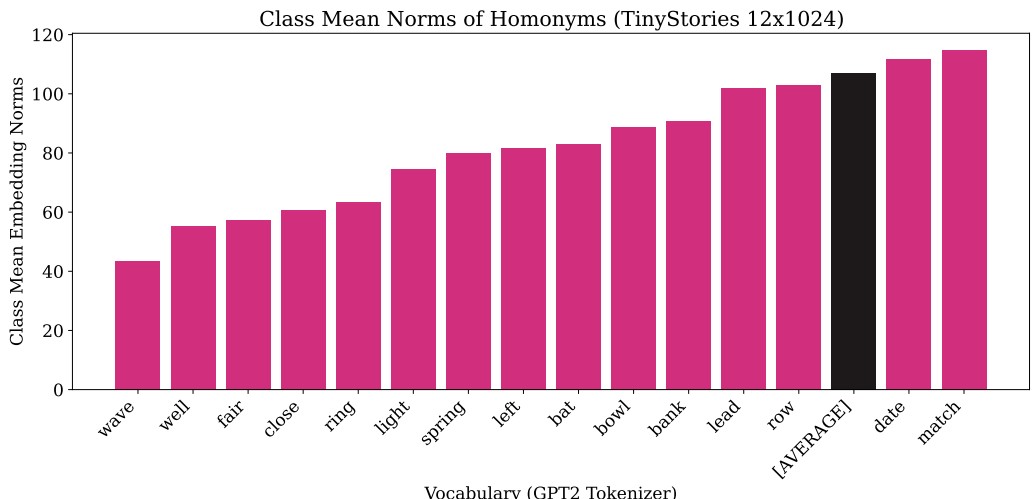

Figure 32: Under `TinyStories-12x1024_10L`, these fifteen homonyms have much shorter mean embedding norms (i.e. closer to the global centre) than the average token. This is expected since homonyms typically present conflicts and interference.

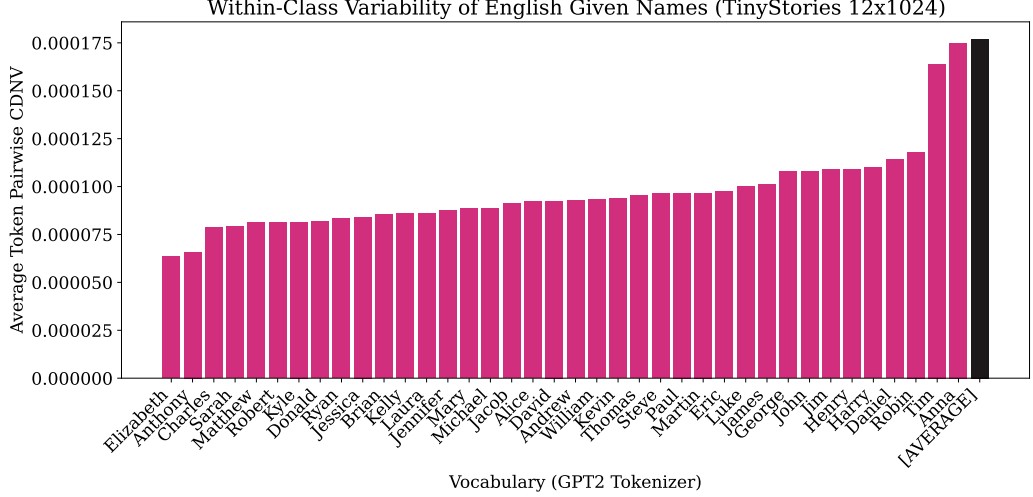

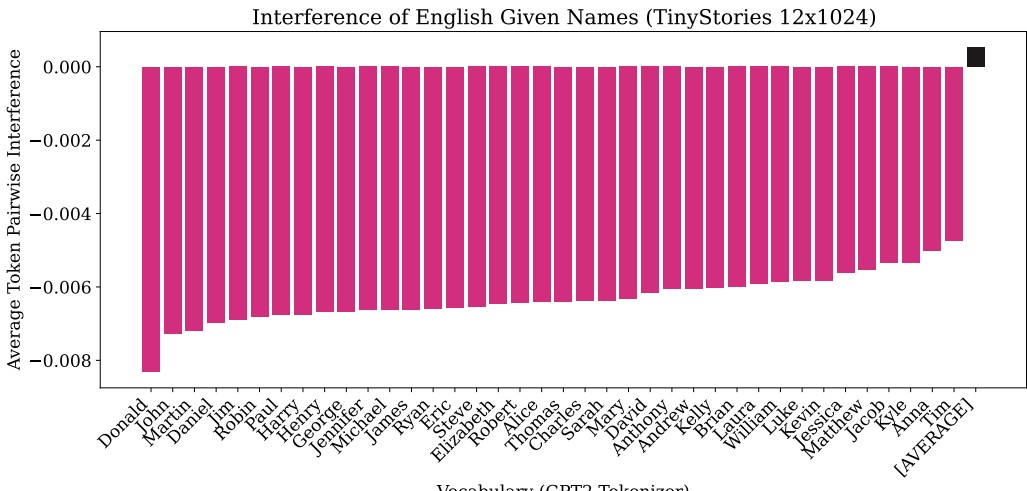

Figure 33: Under `TinyStories-12x1024_10L`, the average within-class variability (top) and interference (bottom) of some English first names were far below those of the average token. This might be because names are distinct and are not typically used in the same contexts as other words (aside from articles). The only names to have CDNV close to that of the average token are "Anna" and "Tim". Note that the positive interference of the average token (right) is not a typo.

