# OpenReview forum: "Linguistic Collapse: Neural Collapse in (Large) Language Models"
_NeurIPS.cc/2024/Conference — NeurIPS 2024 poster_

### Official Review · Reviewer_3JhA · 2024-07-11

**Soundness:** 3
**Presentation:** 3
**Contribution:** 2
**Rating:** 5
**Confidence:** 4

**Summary:**

This paper empirically investigates the emergence of Neural Collapse (NC) properties during the training of causal language models. NC is a phenomenon observed in the top layer of deep-nets trained on one-hot classification problems, where the last-layer class-mean embeddings become equinorm, have maximal angular separation and align with their respective last-layer classifier. However, NC only emerges when: 1) the models are trained beyond the zero training error regime, 2) the number of classes $C$ is larger than the last-layer embedding dimension $d$, 3) the classes are balanced (they have the same number of samples in the training set).

Considering causal language training as the classification of contexts across $C$ words in the vocabulary set, this work explores the extent to which NC emerges in language models, given that none of the three conditions above hold for language models: 1) language models are typically trained for a few epochs, 2) the number of classes $C$ is large, 3) the words frequency in the training set is heavily imbalanced. Since the original NC geometry is not achievable, particularly without condition 2, the authors define *generalized* NC metrics to measure the geometrical properties of the last-layer embeddings and classifiers. They train NeoGPT with varying numbers of layers and width $d$, and analyze the correlation between the (generalized) NC metrics and validation loss across different training regimes.

**Strengths:**

The problem setup is interesting as it attempts to extend previous observations in deep learning to the trending language models, despite their distinct configurations. The scope of the study, its connection to previous works, and the limitations of the setup are well-presented in the paper.

**Weaknesses:**

It is not clear to me whether the suggested NC metrics are suitable for the language setup. For instance, since in a language dataset, a given fixed context might be followed by different next tokens, a model that is accurate in predicting the labels on the training set cannot achieve near-zero NC1. Or regarding UNC3, see question 4 below.

**Questions:**

1. Is your setup only specific to causal language modeling? Particularly, a) Is any part of the formulation or results influenced by the causal/autoregressive training, or is it simply related to the different nature of language datasets in general, such as the large number of classes and the possibility of having multiple labels for a given training sample?, b) Do you expect similar behavior if the experiments were conducted using other language training methods, such as masked language modeling?

2. I don’t understand what you mean by *ambiguous* samples, ``which are not soft- or multi-label’’ (line 128). When a context appears several times in the training set, each time followed by a different next token, this training sample has a soft label, where the label (next token) can take on different values with some non-zero probability for each.

3. (Related to the previous question) What is classification error (line 130) in the context of next-token prediction and what do you mean by "irreducible noise’’? How do you define the error for contexts that appear several times with different next tokens/labels?

4. Intuitively, why do you expect that a model minimizing UNC3 has better generalization? Minimizing NC3 means the classifiers and mean vectors are aligned which is consistent with the classification objective. However, minimizing UNC3 (CoV of the term in Eq (8)), only implies the degree of misalignment between the classifiers $w$ and mean embeddings $\mathbf{\mu}$ is uniform across classes/samples. How does this connect to generalization?

5. Do you observe a decrease in the NC metrics and validation loss as you train the models longer (e.g., 1 vs 10 epochs)?

**Limitations:**

Yes

---

> ### Author Rebuttal · Authors · 2024-08-07
>
> > I don’t understand what you mean by *ambiguous* samples, ``which are not soft- or multi-label’’ (line 128). When a context appears several times in the training set, each time followed by a different next token, this training sample has a soft label, where the label (next token) can take on different values with some non-zero probability for each.
>
> The reviewer is correct that such ambiguous contexts do resemble soft-label data examples, and we will edit the relevant paragraph in the Related Works to reflect this.
>
> > It is not clear to me whether the suggested NC metrics are suitable for the language setup. For instance, since in a language dataset, a given fixed context might be followed by different next tokens, a model that is accurate in predicting the labels on the training set cannot achieve near-zero NC1. Or regarding UNC3, see question 4 below.
>
> We agree that such ambiguous (soft-label) samples such as “Once upon a time ___.” do exist. However, we note that such contexts are more often shorter subsequences typically found in the beginnings of sequences. In longer sequences, the probability of having two identical contexts followed by a different word is diminishingly small. Additionally, longer contexts significantly outnumber the shorter sequences in the training dataset, especially when data and models work with ever longer contexts; this is progressively true in models with increasingly long context windows, such as Llama 3.1 (https://ai.meta.com/research/publications/the-llama-3-herd-of-models/). In the abstract, shorter (ambiguous) contexts could even be excluded or relegated as outliers in the data. Therefore, we conclude that it is likely not an issue overall.
>
> On the other hand, we believe it would still be interesting to extend our work to incorporate soft labels and multi-label approaches. This could be explored using recent advancements in multi-label neural collapse (https://arxiv.org/abs/2310.15903) and mixup neural collapse (https://arxiv.org/abs/2402.06171). We hope that our work serves as a strong first step in modeling NC at the next-token level.
>
> > Is your setup only specific to causal language modeling? Particularly, a) Is any part of the formulation or results influenced by the causal/autoregressive training, or is it simply related to the different nature of language datasets in general, such as the large number of classes and the possibility of having multiple labels for a given training sample?, b) Do you expect similar behavior if the experiments were conducted using other language training methods, such as masked language modeling?
>
> We appreciate the questions. We don’t believe our setup is strictly specific to causal language modeling.
>
> 1. The adverse conditions we list in our Introduction are mostly due to the imbalanced nature of tokens in natural language data. Intuitively, they would apply beyond autoregressive modeling.
> 2. We expect to see similar results in masked language modeling. However, it may depend on which tokens are frequently masked in the data (if a specific scheme is used). We hypothesize that one could conduct experiments with similar results on bidirectional encoders such as BERT or T5 (or their derivatives) but we imagine they’d present additional challenges such as lower token sample efficiency of the MLM paradigm, or the simultaneous prediction or fill-in of multiple dependent/correlated token predictions.
>
> > (Related to the previous question) What is classification error (line 130) in the context of next-token prediction and what do you mean by "irreducible noise’’? How do you define the error for contexts that appear several times with different next tokens/labels?
>
> The loss is token cross-entropy and the error we use is the average mis-classification rate over all tokens. We do not treat ambiguous samples differently and instead defer to the diminishing proportion argument previously made regarding soft-labels. Irreducible noise refers to some minimum loss and error that can't be reduced by more scaling simply because of soft-label contexts.
>
> > Intuitively, why do you expect that a model minimizing UNC3 has better generalization? Minimizing NC3 means the classifiers and mean vectors are aligned which is consistent with the classification objective. However, minimizing UNC3 (CoV of the term in Eq (8)), only implies the degree of misalignment between the classifiers w and mean embeddings μ is uniform across classes/samples. How does this connect to generalization?
>
> Our initial manuscript didn't include our reasoning behind UNC3 and we neglected to provide it in our Appendix, so we thank the reviewer for raising this oversight.
>
> The goal of (self-)duality is to minimize the angles between mean vectors and their corresponding classifiers. This can be measured through the expectation (over class or token pairs) of the squared angle between vectors: $\mathbb E[\theta^2]$.
>
> $$\mathbb E[\theta^2] = \mathbb E[\theta]^2 + \text{Var}[\theta]$$
>
> NC3 measures the average angle $\mathbb E[(\theta)]$, whereas our newly introduced UNC3 measures the variances in the angles $\text{Var}[\theta]$. Achieving duality ultimately requires minimizing both terms.
>
> Once again, we appreciate this comment and will add a short discussion on duality decomposition in $\S$ 3.6 and $\S$ 4.6.
>
> > Do you observe a decrease in the NC metrics and validation loss as you train the models longer (e.g., 1 vs 10 epochs)?
>
> Yes. Appendices D, E, F, G, H, I, J, L, and M all show trends towards NC across training (left-to-right) in sufficiently large models. We show more explicit scatter plots (red = more parameters) in the 1-page PDF supplied in the general rebuttal. That PDF also includes validation loss with respect to training. We will also add these figures to the Appendix of our main manuscript.
>
> ---
>
> Overall, we appreciate the constructive comments of the reviewer. Should our response address the concerns raised, would you consider raising the score? Thank you.

---

> > ### Comment · Reviewer_3JhA · 2024-08-11
> >
> > Thanks for your response.
> >
> > Thanks for clarifying NC3. The intuition appears to rely on both NC3 and UNC3 being correlated with generalization, but Figs. 3, 19, and 21 show negligible/no correlation for NC3. This makes arguments like the one in line 255 somewhat misleading. Defining a metric that correlates with generalization doesn't make it a better replacement for an NC metric that (based on your arguments) is also likely to be tied to generalization. I recommend clarifying this issue in the updated manuscript.
> >
> > Overall, I still feel like the technical novelties for adequately addressing the intricacies of language setups are limited in this paper. However, I also acknowledge that the paper, along with its extensive experiments, can motivate further investigations into extending the NC literature to language tasks. Thus, I will slightly increase my score

---

### Official Review · Reviewer_XbAS · 2024-07-12

**Soundness:** 3
**Presentation:** 2
**Contribution:** 2
**Rating:** 5
**Confidence:** 4

**Summary:**

This work focuses on studying the Neural Collapse phenomenon in the context of language model training. The author first introduces the original NC properties, explaining how such metrics may not apply to the case of LLM training given 1) the ambiguity of language next token prediction, 2) large number of possible tokens , 3) token imbalance and 4) under-parametrization or undertraining of LLM models.

Having established this, the authors introduce new metrics based on the NC ones. Further, the authors analyze the relationship between these metrics and validation loss on a range of models trained on TinyStories dataset. In this regard, 1) collapse of token representation ($ NC_1 $), 2) Hyper-spherical uniformity which represents maximal separation ($ G-NC_2 $), 3) alignment between token feature mean and token classifier ($ U-NC_3 $) and 4) NCC accuracy of features ($ NC4 $) have positive (and stronger) correlation with validation loss.

**Strengths:**

The connection established between Neural Collapse and LLM training is an interesting point. The introduction of the new NC metrics are not only viable for LLMs but also for other classification and training regimes that suffer from similar problems as the language domain.


Further the experiments on TInyStories are performed on a large number of models to establish a connection between the metrics and validation loss and the results could be used to develop evaluation metrics for LLM training.

**Weaknesses:**

1) **(Major)** I appreciate the extensive experimental work on establishing a connection between the proposed NC metrics and validation accuracy; however, I believe the coefficient of determinations provided in Table 1 suggest a low correlation between NC properties and validation loss. While I see the higher correlation for NC1 and NC4 in Fig 1, I’m having a hard time convincing myself that other factors such as model size and architecture don’t play a role in improving the loss.
2) **(Major)** While I understand the attempt at making the autoregressive next token prediction of LLMs comparable to a simple classification setup, I do not think I agree with the “not soft label, not multi label” argument. As the authors suggest, the language format is rather ambiguous. Considering a simple next word prediction, “I went to the ___ “ can be followed by a variety of valid words. Same applies to token baked predictions. I believe not having a mechanism to account for this and dismissing the multi-lableness or soft-lablenss of language removed much of the uniqueness and difficulty of dealing with LLM training.
3) **(Major)** Considering how languages themselves have an imbalanced nature, I wonder whether the suggested ETF of hypersphere features and weights geometric structure are actually optimal for LLMs. As an example, in the case where for a CIFAR10 classifier, we train the model under artificial imbalance, I would understand why symmetric features could potentially help improve test accuracy given the balanced nature of the validation set. However, I have a hard time being convinced the same is true in language.
4) **(Minor)** I find it hard at times to follow the paper’s ideas. Particularly with regards to section 4, I believe a bit more structure or separation of ideas could help readers better digest the conclusions.

**Questions:**

1) Is it true that NC properties are attributed to better generalization for classification ? Could the authors please provide some references.
2) Have the authors considered using smaller or simpler language datasets to combat the problems with large numbers of classes or have models be trained for a longer time in order to see better convergence results for the NC metrics as illustrated in the appendix ?

---

> ### Author Rebuttal · Authors · 2024-08-07
>
> > **(Major)** I appreciate the extensive experimental work on establishing a connection between the proposed NC metrics and validation accuracy; however, I believe the coefficient of determinations provided in Table 1 suggests a low correlation between NC properties and validation loss. While I see the higher correlation for NC1 and NC4 in Fig 1, I’m having a hard time convincing myself that other factors such as model size and architecture don’t play a role in improving the loss.
>
> To clarify, we don't claim that size and training don't contribute to reducing the loss. Our principle observation is that scaling improves both performance (as expected) and NC, and that the two are associated (shown in Figure 1).
>
> To address the concern that scaling alone is the confounding factor, we conducted a seed sweep ($\S$ 4.5) to test an independent correlation. Table 1 shows that most correlations between NC and generalization are statistically significant. Equinormness is not correlated but only accounts for some of NC, a property that is superseded by GNC2 anyway. The only real exception is therefore UNC3, which seems to be entirely confounded by model scale. (Although this does highlight the effect of NC3 on performance independent of scale.)
>
> > **(Major)** While I understand the attempt at making the autoregressive next token prediction of LLMs comparable to a simple classification setup, I do not think I agree with the “not soft label, not multi label” argument. As the authors suggest, the language format is rather ambiguous. Considering a simple next word prediction, “I went to the ___ “ can be followed by a variety of valid words. Same applies to token baked predictions. I believe not having a mechanism to account for this and dismissing the multi-lableness or soft-lableness of language removed much of the uniqueness and difficulty of dealing with LLM training.
>
> Thanks for the detailed comment. Firstly, we agree with the reviewer that there are indeed ambiguous samples, particularly with short sentences, such as the example “I went to the ___.” These cases can resemble soft label data examples, and we will edit the relevant paragraph in the Related Works to reflect this.
>
> We note however that ambiguous contexts are typically shorter and found at the beginning of sequences. For longer sequences, the probability of two identical contexts followed by a different word is low. Additionally, longer contexts outnumber shorter ones in the training data, especially as LLMs (such as Llama 3.1) handle longer contexts. Shorter contexts could even be excluded or treated as outliers. Thus, we conclude that these anomalies likely represent a minority of the data and aren't a significant issue overall. Nonetheless, we take the first step in NC at the next-token level, and future work might incorporate multi-label (arXiv:2310.15903) and mixup (arXiv:2402.06171) NC.
>
> > **(Major)** Considering how languages themselves have an imbalanced nature, I wonder whether the suggested ETF of hypersphere features and weights geometric structure are actually optimal for LLMs. As an example, in the case where for a CIFAR10 classifier, we train the model under artificial imbalance, I would understand why symmetric features could potentially help improve test accuracy given the balanced nature of the validation set. However, I have a hard time being convinced the same is true in language.
>
> Our primary objective is to investigate the structures that form in standard LLM training. We recognize that the convergence geometry in current LLMs may not be optimal, likely due to using CE loss (adapted for balanced data) on an imbalanced dataset. We don't not claim the simplex ETF or hypersphere are optimal; rather, we focus on characterizing the patterns that emerge at scale in relation to baseline geometries. However, recent works have proposed methods for addressing data imbalance (arXiv:2301.00437v5, arXiv:2301.01100, and MLR: Liu et al. (2023)). We hope our insights may lead to adaptations of such methods to optimize these structures that address the imbalanced nature of language
>
> > **(Minor)** I find it hard at times to follow the paper’s ideas. Particularly with regards to section 4, I believe a bit more structure or separation of ideas could help readers better digest the conclusions.
>
> We sympathize with this sentiment. We'll move 4.5 through 4.7 (including Table 1) into a separate section.
>
> > Is it true that NC properties are attributed to better generalization for classification? Could the authors please provide some references?
>
> The original NC showed some relationship to generalization. In particular, we highlight the results in transfer learning (arXiv: 2112.15121) and a generalization bound that can estimate test performance (arXiv: 2202.09028).
>
> We aren't aware of other works that explicitly studied the relationship between NC and generalization. However, our results appear to reinforce the generalization narrative as we observe correlations long before the terminal phase of training, some of which are independent of scale. We hope to inspire future works that might definitively answer this open question.
>
> > Have the authors considered using smaller or simpler language real datasets to combat the problems with large numbers of classes or have models be trained for a longer time to see better convergence results for the NC metrics as illustrated in the appendix?
>
> Thank you for your insightful suggestion. We considered using smaller or simpler language data to potentially improve convergence. However, we focus on large-scale language modeling scenarios to ensure our findings are robust and reflective of contemporary practices. This would render our results applicable and generalizable to real-world applications. The synthetic nature of TinyStories is a caveat to our study (as the reviewer implies), but we provide a lengthy explanation for its use over real data in the general rebuttal above.

---

> > ### Comment · Reviewer_XbAS · 2024-08-13
> >
> > I would like to thank the authors for the elaborate response. With regards to the the multi-label context I understand the arguments regarding the impact of context lenght and I appreciate the explanation. I would like this point to be further clarifier in future revisions. I would further like to encourage evaluation metrics other than test loss to validate the relationship between NC properties and improving LLM quality. Having said this, I would like to increase my score to a 5.

---

### Official Review · Reviewer_Nr3s · 2024-07-12

**Soundness:** 4
**Presentation:** 4
**Contribution:** 4
**Rating:** 8
**Confidence:** 3

**Summary:**

The final layers of classifiers show a property called neural collapse (NC), which are seen as beneficial to model performance. The authors study it’s appearance in causal language models (CLMs), and point out that CLMs do not respect those conditions (CLM are trained on noisy, unbalanced data, made on more tokens than the models have dimensions, and training is stopped before loss reaches 0).
They then provide a theoretical framework to adapt the NC properties to CLMs (arguing for two relaxations from past works: measuring hyperspherical uniformity, and uniform duality), and consider show that despite expectations, evidence of NC exists, and is stronger for better performing models.

**Strengths:**

Provided measured and well formulated novel results regarding the impact of scale, training, number of parameters on the different components of generalisation. While a highly technical read which required a lot of concentration, all the required information for understanding the concepts are introduced. Conclusions are measured and entirely based on evidence from theory and empirical evidence, yet discussion manages to highlight the implication of such research. Exciting mathematical tools I hope to use in the future.

**Weaknesses:**

Apart from evaluation loss, model performance metrics from commonly used LLM benchmarks are not provided, making high level comprehension of the technical observations more complicated.

**Questions:**

considering CLMs to be classifiers is technically sound, nonetheless unlike with classification tokens can mean different things (ex:homonyms) these might also explain divergence from the classic NC model. Did you observe any such groupings or effects?

**Limitations:**

Limitations are to my knowledge well adressed

---

> ### Author Rebuttal · Authors · 2024-08-07
>
> > Apart from evaluation loss, model performance metrics from commonly used LLM benchmarks are not provided, making high level comprehension of the technical observations more complicated.
>
> LLM researchers are indeed interested in performance metrics beyond the cross-entropy (CE) evaluation loss that we use. However, as our work concerns the measurement of NC under model pre-training only, other benchmarks (particularly those on downstream tasks) would not necessarily be appropriate as they measure specific capabilities rather than generic stochastic token prediction; therefore, they would be out-of-scope for our work.
>
> Furthermore, according to several recent works on LLM evaluations (https://arxiv.org/abs/2403.15796, https://arxiv.org/abs/2404.09937, https://arxiv.org/abs/2407.06645), downstream capabilities of LLMs are roughly correlated with their abilities to compress their pre-training data. Based on these findings, we find that CE loss is the most sensible metric with which to measure generalization.
>
> We thank the reviewer for raising this question as we see fit to include a short discussion on this and potential future directions towards the end of our manuscript (citing the aforementioned works).
>
> > Considering CLMs to be classifiers is technically sound, nonetheless unlike with classification tokens can mean different things (ex:homonyms) these might also explain divergence from the classic NC model. Did you observe any such groupings or effects?
>
> Appreciate the question! We inspected token-wise NC values in our largest and most-trained model (12-layer d=1024, 10 epochs). Based on your suggestion, we chose 15 homonyms and found that most have much shorter mean vector norms (meaning they’re closer to the global center) than the average token. This makes sense as homonyms present conflicts and interference.
>
> | GPT-Neo Token ID | Token Text | Scaled Norm |
> | ---------------- | ---------- | ----------- |
> | 808              | row        | 102.9727    |
> | 1806             | ring       | 63.4789     |
> | 2971             | light      | 74.4070     |
> | 4053             | well       | 55.3457     |
> | 4475             | date       | 111.7086    |
> | 8664             | bat        | 83.0259     |
> | 9464             | left       | 81.7467     |
> | 15699            | match      | 114.7236    |
> | 16469            | spring     | 80.0322     |
> | 17796            | bank       | 90.8248     |
> | 19204            | wave       | 43.5028     |
> | 19836            | close      | 60.6004     |
> | 22043            | fair       | 57.2620     |
> | 28230            | lead       | 102.0583    |
> | 36859            | bowl       | 88.7034     |
> | AVERAGE          | AVERAGE    | 106.8762    |
>
> We also observed that the individual variability and interference of some English first names all other tokens were far below the average. This is also intuitive as names are distinct from one another and aren't typically used in the same contexts as other words (aside from articles).
>
> | GPT-Neo Token ID | Token Text | CDNV         | Interference |
> | ---------------- | ---------- | ------------ | ------------ |
> | 7371             | Donald     | 8.176141e-05 | -0.008308    |
> | 7554             | John       | 0.000108     | -0.007270    |
> | 11006            | David      | 9.249406e-05 | -0.006167    |
> | 12041            | Paul       | 9.661650e-05 | -0.006768    |
> | 13256            | Michael    | 8.857495e-05 | -0.006625    |
> | 14731            | James      | 0.000101     | -0.006610    |
> | 14967            | Tim        | 0.000164     | -0.004741    |
> | 17121            | William    | 9.332073e-05 | -0.005867    |
> | 18050            | Jim        | 0.000109     | -0.006896    |
> | 18308            | Harry      | 0.000110     | -0.006749    |
> | 19156            | Robert     | 8.133316e-05 | -0.006438    |
> | 19206            | Steve      | 9.622102e-05 | -0.006541    |
> | 19962            | Daniel     | 0.000114     | -0.006963    |
> | 20191            | George     | 0.000108     | -0.006674    |
> | 20508            | Andrew     | 9.280709e-05 | -0.006036    |
> | 21868            | Ryan       | 8.332534e-05 | -0.006594    |
> | 22405            | Thomas     | 9.526886e-05 | -0.006400    |
> | 23865            | Kevin      | 9.392182e-05 | -0.005817    |
> | 24119            | Mary       | 8.846451e-05 | -0.006322    |
> | 24761            | Brian      | 8.554082e-05 | -0.006003    |
> | 24778            | Martin     | 9.663536e-05 | -0.007183    |
> | 25004            | Eric       | 9.731254e-05 | -0.006565    |
> | 25372            | Matthew    | 8.109680e-05 | -0.005544    |
> | 28711            | Charles    | 7.878443e-05 | -0.006371    |
> | 29284            | Sarah      | 7.926711e-05 | -0.006366    |
> | 30730            | Luke       | 0.000100     | -0.005830    |
> | 31160            | Anna       | 0.000175     | -0.005005    |
> | 32476            | Henry      | 0.000109     | -0.006685    |
> | 32697            | Anthony    | 6.594844e-05 | -0.006055    |
> | 34831            | Kelly      | 8.580151e-05 | -0.006017    |
> | 40656            | Robin      | 0.000118     | -0.006808    |
> | 42516            | Kyle       | 8.159300e-05 | -0.005333    |
> | 43187            | Jennifer   | 8.750914e-05 | -0.006629    |
> | 43568            | Elizabeth  | 6.347037e-05 | -0.006462    |
> | 43687            | Laura      | 8.584216e-05 | -0.005922    |
> | 44484            | Alice      | 9.208282e-05 | -0.006416    |
> | 45572            | Jessica    | 8.405318e-05 | -0.005614    |
> | 46751            | Jacob      | 9.124901e-05 | -0.005349    |
> | AVERAGE          | AVERAGE    | 0.000177     | 0.000519     |
>
> There are probably thousands of groupings or linguistic relationships one could observe in the NC measurements, so we’ll leave such interpretability to future applications.
>
> ---
>
> We thank the reviewer for the thought-provoking questions. We hope our responses adequately address them and warrant a slightly higher score.

---

> > ### Comment · Reviewer_Nr3s · 2024-08-09
> >
> > Thank you for your response, those additional results kind of also fit in with my other question on common LLM benchmarks, they provide examples of use cases of the mathematical framework. I maintain my opinion that this is an excellent work, and maintain the high score.

---

### Official Review · Reviewer_NZtr · 2024-07-13

**Soundness:** 4
**Presentation:** 4
**Contribution:** 3
**Rating:** 7
**Confidence:** 4

**Summary:**

This paper investigates neural collapse (NC) -- properties of the penultimate feature representation of DNN -- in causal language models (CLM). Previous NC studies have primarily focused on classification problems with balanced classes with few labels compared to the feature dimensionality. This paper finds that NC properties in CLM are correlated with generalisation. This finding is based on extensive experimental evaluations on the TinyStories dataset.

**Strengths:**

* The NC analysis is interesting and it's noteworthy that properties of NC appear to correlate with generalisation in CLMs, which differ in important ways from other classification settings where NC has been studied.

* The empirical results seem to paint a clear trend in terms of NC and generalisation.

* The presentation is generally quite good, with contributions and significance clearly spelled out.

**Weaknesses:**

* I’m somewhat unclear on the motivation for using TinyStories, a purely synthetic dataset. While I understand the benefit of using a small dataset to ease the computational burden of the experiments, I would have liked to see some further experiments on real data confirming the observed trends. For example, you could sample a small dataset of human-written stories.

* I would have liked to see more practical guidance on the value of the proposed metrics. For example, it’s not clear if they provide any value in terms of assessing model fit beyond what a held-out validation would (more cheaply) provide.

* In several places, the wording is unusual / confusing. For example, L47-L48 says that "[...] LLM learn to model aleatoric
uncertainty, which can be viewed as stochastic token prediction," which is a peculiar/wordy way to describe a standard MLE procedure.

* While the experiments are fairly comprehensive, the technical novelty is relatively low. The paper mostly consists of methods from prior works being applied to a new classification setting.

**Questions:**

See "Weaknesses."

**Limitations:**

Yes.

---

> ### Author Rebuttal · Authors · 2024-08-07
>
> > I’m somewhat unclear on the motivation for using TinyStories, a purely synthetic dataset. While I understand the benefit of using a small dataset to ease the computational burden of the experiments, I would have liked to see some further experiments on real data confirming the observed trends. For example, you could sample a small dataset of human-written stories.
>
> This crucial question arose during our research, as using a synthetic dataset may limit the generality of our results to human-written data (see our lengthy explanation in the general rebuttal). We acknowledge the importance of validating our findings with real data and are aware of follow-up efforts to scale our experiments with appropriately sized models and datasets.
>
> > I would have liked to see more practical guidance on the value of the proposed metrics. For example, it’s not clear if they provide any value in terms of assessing model fit beyond what a held-out validation would (more cheaply) provide.
>
> NC has been studied in and in relation to many areas of machine learning. We defer the vast related works and applications to the Related Works and Discussion sections of our manuscript, while our response here will therefore be tailored towards practical consequences and applications of studying NC in CLMs.
>
> 1. Better optimization strategies: LLMs are typically pre-trained with CE loss in teacher-forcing. It stands to reason that works listed in the “Learning to collapse” Discussion paragraph can be applied or inspire better objective functions towards potentially better geometric configurations for language models. Hierarchical structures like those presented by Liang & Davis (2023). The implication is that future LLMs could be trained in fewer steps or to better performance.
> 2. Interpretability of contextual token-wise behaviors: our measurements provide insight into behaviors of individual token embeddings. Token-wise NC3 can reveal how confident an LLM is about certain tokens in their top-layer classifiers. For example, the number and density of clusters for a particular token can shed light into its various meanings and uses. This would be particularly useful as LLMs adapt to ever-evolving use of language and further expansion into non-English domains.
> 3. Interpretability of pair-wise interactions: our measurements primarily center around pair-wise relationships between tokens in a vocabulary. The pair-wise arrangements are critically important to modeling interference between tokens because their context vectors live on a lower-dimensional hypersphere. For example, you can tell how related or interchangeable two words are based on their noise and interference (NC1,2), or how antithetical or unrelated they are based on orthogonality (NC2).
> 4. Interpretability for fairness: natural language is inherently imbalanced. Individual or pair-wise analyses of representations can reveal inequalities within the vocabulary or across topics. These insights can guide researchers and engineers to address inequities in a targeted and measurable way.
> 5. Interpretability for ethics: if there are concerns about biases or safety issues in general language modeling, NC analysis can aid researchers in interpreting how LLMs could pose risks to users, and how to safely mitigate risks without degrading model quality.
> 6. Continual learning: [19, 20, 21] cited studied NC in (task-)incremental learning problems and provided insights or improvements. Since an LLM (especially a foundation model) is to continually learn language patterns and develop capabilities, NC can provide carefully structured configurations that allow for graceful lifelong learning.
>
> Of course it remains to be seen in exactly what form or role NC will play in these areas of further studies. It is also probable there are benefits of NC analysis beyond our knowledge. Our work here is to take the first step in extending the body of NC work into the more irregular and demanding settings of causal language modeling.
>
> We are grateful to the reviewer for probing this response. We will further expand our Significance and Discussion (sub)sections to explicate these practical guidelines.
>
> > In several places, the wording is unusual / confusing. For example, L47-L48 says that "[...] LLM learn to model aleatoric uncertainty, which can be viewed as stochastic token prediction," which is a peculiar/wordy way to describe a standard MLE procedure.
>
> We agree with the reviewer and will revise L47-48 to “[…] learn to stochastically predict tokens to generate text.” We also identified the following opportunities for rewording or clarification:
>
> 1. L107-108 will be adjusted: “[…] classes number in the tens of thousands.”
> 2. L128 will be clarified such that ambiguous contexts *are* “soft-label” samples.
> 3. L230-231 will be rewritten: “These noise reductions are associated with generalization (Fig. 1, left, “N C1”); this relationship grows stronger with model size.”
>
> Should the reviewer feel there are more instances that we did not cover, we will happily review further during the discussion period. And of course, we will perform more passes on this paper to improve the writing for future revisions.
>
> > While the experiments are fairly comprehensive, the technical novelty is relatively low. The paper mostly consists of methods from prior works being applied to a new classification setting.
>
> This is a fair point, that we mostly applied previous expressions and methods to the causal language modeling setting. Our contribution focuses on applying the canonical NC framework to more adverse conditions (larger and imbalanced vocabulary, ambiguous contexts, undertraining), the effort of which is reflected in our work. We hope that this first attempt at analyzing NC in this area inspires further work that better adapts to the ever-changing landscape.
>
> ---
>
> We are grateful for the helpful comments. Should our response warrant it, we would greatly appreciate the reviewer raising the score.

---

> > ### Comment · Reviewer_NZtr · 2024-08-11
> >
> > Thanks for the response! I maintain my accept recommendation.

---

### Author Rebuttal · Authors · 2024-08-07

## The Choice of TinyStories

The study of NC in causal language modeling at the token level would be very expensive, so the motivation to use a small dataset is clear. However, most commonly used text datasets such as WikiText, BookCorpus, CommonCrawl, or most subsets from the Pile are much too complex and broad to be effectively compressed by CLMs of the scale that we work with.

WikiText-2 and WikiText-103 present significant drawbacks for our experiments. Both datasets contain a considerable amount of low-quality data that doesn't concentrate on essential linguistic structures such as grammar, vocabulary, facts, and reasoning. WikiText-2 has a similar empirical vocabulary to TinyStories under the GPT-Neo tokenizer (27K vs. 29K) but only has around 34K rows of training data compared to 2.1M in TinyStories. Our small-scale NC experiment on WikiText-2 revealed that the models were very brittle and prone to overfitting. On the other hand, WikiText-103 is comparably sized to TinyStories but utilizes around 44K unique tokens. Our CLMs trained on WikiText-103 struggled to produce coherent sentences, likely due to the excessive breadth and information, as noted by the authors of TinyStories. Beyond these two, we were unable to find any real datasets that both followed established scaling laws (Kaplan et al., Hoffman et al.) for CLMs at our scale and are simple enough to suit the analysis of NC.

This is where the TinyStories dataset becomes invaluable. Their manuscript (Eldan & Li) from last year informs much of our reasoning. This is an excerpt from their manuscript:

> We introduce TinyStories, a synthetic dataset of short stories that only contain words that a typical 3 to 4-year-olds usually understand, generated by GPT-3.5 and GPT-4. We show that TinyStories can be used to train and evaluate LMs that are much smaller than the state-of-the-art models (below 10 million total parameters), or have much simpler architectures (with only one transformer block), yet still **produce fluent and consistent stories with several paragraphs that are diverse and have almost perfect grammar, and demonstrate reasoning capabilities.**

According to its authors, it’s explicitly designed to preserve the essential elements of natural language, such as grammar, vocabulary, facts, and reasoning, while being smaller and more refined in terms of its breadth and diversity. Unlike large corpora that can overwhelm small language models (SLMs) due to their excessive breadth and diversity, TinyStories offers a concentrated dataset that hones in on core linguistic structures and reasoning capabilities. This is evident in its small word vocabulary, consisting of approximately 1500 words that a child would use, and in its 29K empirical vocabulary under the GPT-Neo tokenizer.

Despite its concentrated nature, TinyStories enables models trained on it to produce grammatically correct, factual, and reasonable stories. Additionally, these models can be finetuned on specific instructions found in the TinyStories-Instruct dataset. The authors of TinyStories also demonstrate that their models can creatively produce stories that are dissimilar enough to their training data, indicating a balanced capability for generalization and creativity.

One particular advantage of TinyStories is the small vocabulary relative to total training tokens. The consequence is a reasonable number of classes with higher average token counts. Conveniently, frequency analysis of the overall dataset produced a distribution (Figure 4 in first draft) that is similar to real human language. This is relevant because the ability to measure NC and a CLM’s ability to compress language data into distinct geometries depends partially on the ratios between embedding dimension, vocabulary size, and average token counts. TinyStories provides a good balance for an initial study in this phenomenon.

Additionally, TinyStories has more regular structure as GPT-3.5/4 were instructed to produce children’s stories with certain themes and forms with a limited vocabulary. We believed that this would reduce the amount of clustering noise from the very broad information and structures in real general data, and allow our smaller CLMs to exhibit clear trends towards NC.

Furthermore, TinyStories was created using GPT 3.5/4, which are advanced language models with significantly larger architectures trained on orders of magnitude more tokens, helping minimize the effect of the synthetic nature of the generated dataset. We also considered a possible effect of model collapse as a result of training on synthetic data but Shumailov et al. and some follow-up works suggest that a single iteration of data generation (as generated TinyStories) has a very negligible model collapse.

With all that said, we are deeply grateful to the reviewers who raised this issue. We will include the above explanation at length in our Appendix B and a summary in $\S$ 3.1 where we introduce TinyStories.

### References

- Kaplan et al. (2020): https://arxiv.org/abs/2001.08361
- Hoffman et al. (2022): https://arxiv.org/abs/2203.15556
- Eldan & Li (2023): https://arxiv.org/abs/2305.07759
- Shumailov et al. (2023): https://arxiv.org/abs/2305.17493

---

### Decision · Program_Chairs · 2024-09-25

**Decision:**

Accept (poster)

**Comment:**

The paper explores the phenomenon of neural collapse (NC) in the context of causal language models (CLMs). The authors aim to understand how NC properties, which have been predominantly studied in well-behaved classification settings that are trained on noisy, unbalanced data with a large and often imbalanced vocabulary. The paper applies the concept of NC to a new domain, CLMs, which differs significantly from the classification settings where NC has been traditionally studied, which is novel and valuable. The authors provide a comprehensive empirical evaluation, demonstrating a correlation between the proposed NC metrics and generalization in CLMs. One main concern raised by the reviewers is the use of a synthetic dataset (TinyStories), which may limit the generalizability of the findings to real-world human-written language data.